# Phenomenal Yet Puzzling: Testing Inductive Reasoning Capabilities of Language Models with Hypothesis Refinement

**Linlu Qiu**[1,*] **Liwei Jiang**[2,3], **Ximing Lu**[2,3], **Melanie Sclar**[3], **Valentina Pyatkin**[2,3],
**Chandra Bhagavatula**[2], **Bailin Wang**[1], **Yoon Kim**[1], **Yejin Choi**[2,3], **Nouha Dziri**[2], **Xiang Ren**[2,4]
[1]Massachusetts Institute of Technology, [2]Allen Institute for Artificial Intelligence
[3]University of Washington, [4]University of Southern California
`linluqiu@mit.edu`

## Abstract

The ability to derive underlying principles from a handful of observations and then generalize to novel situations—known as inductive reasoning—is central to human intelligence. Prior work suggests that language models (LMs) often fall short on inductive reasoning, despite achieving impressive success on research benchmarks. In this work, we conduct a systematic study of the inductive reasoning capabilities of LMs through *iterative hypothesis refinement*, a technique that more closely mirrors the human inductive process than standard input-output prompting. Iterative hypothesis refinement employs a three-step process: proposing, selecting, and refining hypotheses in the form of textual rules. By examining the intermediate rules, we observe that LMs are phenomenal *hypothesis proposers* (i.e., generating candidate rules), and when coupled with a (task-specific) symbolic interpreter that is able to systematically filter the proposed set of rules, this hybrid approach achieves strong results across inductive reasoning benchmarks that require inducing causal relations, language-like instructions, and symbolic concepts. However, they also behave as puzzling *inductive reasoners*, showing notable performance gaps between rule induction (i.e., identifying plausible rules) and rule application (i.e., applying proposed rules to instances), suggesting that LMs are proposing hypotheses without being able to actually apply the rules. Through empirical and human analyses, we further reveal several discrepancies between the inductive reasoning processes of LMs and humans, shedding light on both the potentials and limitations of using LMs in inductive reasoning tasks.[1]

## 1 Introduction

*Inductive reasoning*, i.e., the ability to identify common patterns and form high-level abstractions from limited observations, is considered key to human intelligence (Lake et al., 2017; Chollet, 2019). For instance, humans can quickly identify the *generalizable* list operation rule "selecting the first item" based on only a few observations (Figure 1, top). Although the precise cognitive mechanisms behind inductive reasoning remain unknown, one compelling hypothesis in cognitive science posits that humans approach this challenge through an *iterative* process that involves proposing hypotheses, testing them against observations, and refining them accordingly (Heit, 2000; Fränken et al., 2022). Returning to the above example, while the hypothesis "selecting the smallest item" may seem plausible based on the first two examples, applying this rule to the final example reveals the need for refinement, ultimately favoring "selecting the first item" as a more accurate hypothesis.

With the increasing power of state-of-the-art LMs (OpenAI, 2023; Anthropic, 2023), there is growing interest in exploring these models' reasoning capabilities vis-à-vis human inductive reasoning. How are their performances and underlying mechanisms similar to (and contrasted with) those of humans? This work investigates LMs' inductive reasoning capabilities through the lens of iterative

---

*Work done during an internship at Allen Institute for AI.

[1]We release our code at `https://github.com/linlu-qiu/lm-inductive-reasoning`.

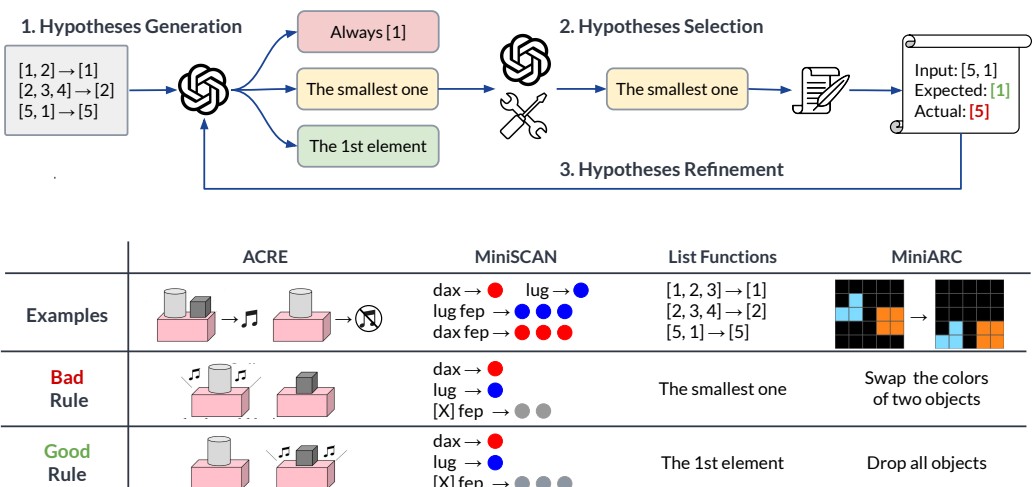

Figure 1: An overview of the iterative hypothesis refinement approach. We generate $N$ hypotheses per iteration and iterate up to the maximum number of iterations $T$ (top). Example instances and representative good and bad rules for each task (bottom).

hypothesis refinement: *hypotheses generation, selection, and refinement*. Specifically, we use an LM to propose a set of free-form or constrained hypotheses based on observations. The proposed hypotheses are then verified against observations via off-the-shelf symbolic interpreters[2], e.g., grammar parsers or code interpreters, which can determine if an hypothesis applies to specific instances. The hypothesis that covers most number of observations is then selected to be further refined by the LM. This process is repeated to induce the final hypothesis.

Results across four distinct tasks, including inducing causal relations (Zhang et al., 2021), language-like compositional instructions (Lake et al., 2019), symbolic operations (Rule, 2020), and visual concepts (Kim et al., 2022b), show that this iterative hypothesis refinement process significantly improves upon standard input-output (IO) prompting. We find that LMs are particularly good at generating candidate rules, and when coupled with a symbolic interpreter that can provide accurate feedback with which to refine hypotheses, this hybrid induction approach is effective.

However, a closer inspection of the refinement pipeline reveals a more nuanced view of the putative inductive reasoning process of LMs. Despite being able to generate plausible candidate rules, LMs display a range of puzzling counterintuitive behaviors. For one, while we might expect humans to be able to apply the rules they propose, we find that LMs are often unable to correctly apply their own proposed rules (§4.1). Moreover, while humans can make robust inductions by abstracting away from small perturbations present in examples (e.g., different representational forms of examples), we observe LMs to be highly brittle in the face of even minor perturbations (§4.2). Finally, a human study reveals that rules induced by LMs generally have different content and form compared to rules generated by humans. LMs often provide verbose descriptions of patterns but fail to leverage pragmatic communication strategies commonly seen in human induction (§4.3).

Our study unveils the paradoxical inductive capabilities of LMs: they are simultaneously *phenomenal hypothesis proposers* and *puzzling inductive reasoners*. Our paper connects to classical approaches for concept induction (Tenenbaum et al., 2011; Ellis et al., 2023, *i.a.*), latent language optimization (Andreas et al., 2018; Mu et al., 2020, *i.a.*), and instruction induction (Honovich et al., 2023). While similar in spirit to recent work on exploring inductive reasoning with LMs (Wang et al., 2023a), our work offers a nuanced exploration of both the potentials and limitations of LMs.

## 2 INDUCTIVE REASONING WITH LMS: EXPERIMENTAL SETUP

We consider the rule induction problem of inferring a function $f : \mathcal{X} \to \mathcal{Y}$ that maps an input $x \in \mathcal{X}$ to an output $y \in \mathcal{Y}$. The rule, $f$, can take various forms, such as mathematical operations, grammar,

---

[2]If the hypothesis is in free-form natural language, we additionally ask an LM to translate it into a specific format, e.g., code, that is interpretable by the symbolic interpreter. See Appendix D for examples.

and even natural language descriptions (see Appendix D for examples). For each task $\tau$, we have a set of examples $\mathcal{D}_\tau$ consisting of input-output pairs $(x, y)$. We further divide $\mathcal{D}_\tau$ into seen examples, $\mathcal{D}_\tau^s$, and unseen examples, $\mathcal{D}_\tau^u$. The goal is to induce the $f$ that best describes $\mathcal{D}_\tau$ using only $\mathcal{D}_\tau^s$. A good rule thus requires a balance between precision and coverage, i.e., it should be simultaneously *expressive* enough to capture $\mathcal{D}_\tau^s$ and *generalizable* to $\mathcal{D}_\tau^u$.

We assess an LM's ability to induce rules through prompting. Let $h \in \Sigma^*$ be a rule generated by an LM, where $\Sigma$ is the LM's vocabulary. Since we cannot directly apply $h$ to $x$ ($h$ is just a piece of text), we make use of an interpreter $I_\tau : \Sigma^* \to \mathcal{F}$ for each task $\tau$ where $\mathcal{F}$ is the space of all functions from $\mathcal{X}$ to $\mathcal{Y}$ (i.e., $f \in \mathcal{F}$). That is, the interpreter $I_\tau$ "compiles" $h$ into a function that can be applied to $x$.[3] The quality of rules is evaluated based on their performance on unseen examples. Given an induced rule $h$ and $n$ unseen examples $\mathcal{D}_\tau^u = \{(x_1, y_1), ..., (x_n, y_n)\}$, we derive outputs $y_i'$ by applying $I_\tau(h)$ to input $x_i$,

$$y_i' = I_\tau(h)(x_i). \tag{1}$$

Although it is ideal to have interpreters that can correctly apply $h$, such perfect interpreters might not always be available. Importantly, interpreters have no access to $\mathcal{D}_\tau^s$, and thus, the rule must contain sufficient information for interpreters to achieve strong performance when applying the rule.

We evaluate the quality of a rule $h$ using accuracy. More formally, for a task $\tau$ containing a set of unseen examples $\mathcal{D}_\tau^u$, we first define the accuracy for this particular task as

$$a_\tau = \frac{1}{|\mathcal{D}_\tau^u|} \sum_{(x,y) \in \mathcal{D}_\tau^u} \mathbb{1}\big[I_\tau(h)(x) = y\big]. \tag{2}$$

Let $\mathcal{T}$ denotes the set of all tasks within a dataset. We define *raw accuracy* $c$ and *task accuracy* $c_t$ as

$$c = \frac{1}{|\mathcal{T}|} \sum_{\tau \in \mathcal{T}} a_\tau \qquad c_t = \frac{1}{|\mathcal{T}|} \sum_{\tau \in \mathcal{T}} \mathbb{1}\big[a_\tau = 1\big]. \tag{3}$$

While raw accuracy is the standard metric used in prior work, task accuracy could better estimate an LM's induction capability: a model should ideally consistently solve examples within a task. We use GPT-4 (`gpt-4-0613`; OpenAI, 2023) for all experiments and analyses. We include additional results of other models, including GPT-3.5 (`gpt-3.5-turbo-0613`), Claude-2 (Anthropic, 2023), and LLaMA2-70B (Touvron et al., 2023) in Appendix B.

## 2.1 Iterative Hypothesis Refinement

We consider an iterative approach to induce rules from LMs. We use LMs to propose a set of rules (i.e., hypotheses). We then select the best rule based on scores calculated using the interpreter function. We provide feedback to LMs for further refinement. See Figure 1 for an overview.

Specifically, given $k$ exemplars $\mathcal{D}_\tau^s = \{(x_1, y_1), ..., (x_k, y_k)\}$, at iteration $t$, we sample $N$ hypotheses of rules, $H^t = \{h_1^t, ..., h_N^t\}$, from a prompted LM,

$$h^t \sim P_{\text{LM}}\big(\cdot \mid d^{t-1}, x_1, y_1, ..., x_k, y_k\big), \tag{4}$$

where $d^{t-1}$ is the feedback from previous iterations and which is set to be an empty string at the initial iteration. Each hypothesis is re-ranked based on a scoring function $s(h, \mathcal{D}_\tau^s)$. We use accuracy over seen examples as the scoring function,

$$s(h, \mathcal{D}_\tau^s) = \frac{1}{|\mathcal{D}_\tau^s|} \sum_{(x,y) \in \mathcal{D}_\tau^s} \mathbb{1}\big[I_\tau(h)(x) = y\big]. \tag{5}$$

The best hypothesis is selected via,

$$h^{t^*} = \underset{h' \in H^t}{\arg\max}\, s(h', \mathcal{D}_\tau^s). \tag{6}$$

We then obtain feedback $d^t$ by passing the best hypothesis to a template-based feedback generator. The feedback $d^t$ is a concatenation of exemplars with incorrect predictions, formatted as *input*, *expected output*, and *tentative output*. The iteration stops if the interpreter produces correct outputs for all exemplars using the current hypothesis or if the maximum iteration $T$ is reached. In all experiments, we consider a combination of maximum number of iterations $T \in \{1, 3\}$ and number of hypotheses per iteration $N \in \{1, 5\}$. We use greedy decoding when generating a single hypothesis and set the temperature to 0.7 when generating multiple hypotheses following Wang et al. (2023b).

---

[3]For example, if $h$ is the string representation of a Python function, $I_\tau(h)$ can be the actual Python function. Note that the same rule $h$ could be applied differently by different interpreters.

## 2.2 DATASETS

The above framework requires three specifications: the rule function $f$, the representation (i.e., the format and content) of $h$, and the interpreter $I$. We evaluate on 4 datasets (where each dataset consists of multiple tasks) and formulate these specifications as follows (see examples in Figure 1). We show the full dataset details in Appendix A.

**ACRE.** The Abstract Causal REasoning (ACRE; Zhang et al., 2021) is a diagnostic dataset designed to evaluate causal induction ability. It requires identifying a set of "Blicket" objects that will trigger a special machine. We can view $f$ as an indicator function s.t. $f(x; B) = \mathbb{1}[B \cap x]$ where $B$ is the set of Blickets and $x$ is the presented objects. We constrain $h$ to classify each object into one of three categories: a Blicket, not a Blicket, or undetermined. $I(h)(x)$ is thus a deterministic function that checks the intersections between current objects and predicted Blickets.

**MiniSCAN.** Lake et al. (2019) developed a sequence-to-sequence task with only 14 training examples to measure few-shot concept learning ability. We refer to this as MiniSCAN following Nye et al. (2020).[4] Similar to SCAN (Lake & Baroni, 2018), MiniSCAN requires translating an input command $x$ to an output action sequence $y$. We consider $f$ as a set of grammar rules that map the input symbols to the corresponding meaning representations. We use a quasi-synchronous context free grammar (Smith & Eisner, 2006) as our formalism for $h$ and use a parser as our interpreter $I(h)$.

**List Functions.** The List Functions dataset (Rule, 2020) is designed to evaluate human and machine concept learning ability. It requires identifying a function that maps the input list to its corresponding output list. Here $f$ is a primitive or compositional list operation. We represent $h$ as natural language description and implement the interpreter $I$ using a two-stage process. First, we ask an LM to translate the natural language hypothesis $h$ into a Python program. Then we execute this program to produce the corresponding outputs for given inputs.[5]

**MiniARC.** The Abstract Reasoning Corpus (ARC; Chollet, 2019) and its variants (Kim et al., 2022b; Acquaviva et al., 2022; Xu et al., 2023b; Moskvichev et al., 2023) aim to evaluate visual abstract reasoning over broad concepts. The $f$ here involves a transformation between input and output 2D grids, such as moving an object or swapping colors. We use natural language hypotheses $h$ and similarly interpret the hypotheses using a Python interpreter. Given the extensive grid size of the original ARC tasks and the limited context length of LMs, we consider MiniARC (Kim et al., 2022b), a 5x5 compact version of the ARC.

## 3 LMS ARE PHENOMENAL HYPOTHESIS **PROPOSERS**

**Main Results.** We compare hypothesis refinement with standard input-output (IO) prompting, self-consistency prompting (SC; Wang et al., 2023b), and self-refine (SR; Madaan et al., 2023).[6] SC samples multiple outputs and selects the most consistent one by taking a majority vote. SR uses the same LM as an interpreter and provides feedback to itself, and is a "pure LM" baseline that does not utilize a symbolic interpreter. The results are shown in Table 1 (see Appendix C for existing human performance). Iterative hypothesis refinement achieves the strongest performance on 3 out of 4 datasets, demonstrating the effectiveness of this approach. However, it lags behind the baselines on raw accuracy of MiniARC, potentially because some tasks in MiniARC are heavily dependent on pattern matching, for which IO prompting might be more effective (Mirchandani et al., 2023). Additionally, due to the limited visual understanding capabilities inherent in text-only models, the performance on MiniARC is still far from optimal for all methods, in comparison to other datasets.[7]

Similar to prior work (Chen et al., 2023a; Olausson et al., 2023; Peng et al., 2023, *i.a.*), sampling more hypotheses and using iterative refinement with external feedback significantly boost LM per-

---

[4]This task is also sometimes referred to as "Colors" (Akyurek & Andreas, 2021; Patel et al., 2022).

[5]Although this method might introduce potential errors due to mistranslations between natural language and code, in practice, we qualitatively examine the programs and find that LMs can often generate programs that faithfully represent the natural language hypotheses.

[6]These cannot be directly compared with our method, as hypothesis refinement is augmented with symbolic interpreters. We include them as baselines as they are standard approaches used in existing studies.

[7]We also experimented with a multimodal model on MiniARC but found that it performs worse than text-only models. See Appendix B.2 for details.

Table 1: Iterative hypothesis refinement results. $T$ refers to the maximum number of iterations. $N$ refers to the number of candidate hypotheses per iteration.

| Method | Raw Accuracy | | | | Task Accuracy | | | |
|---|---|---|---|---|---|---|---|---|
| | ACRE | MiniSCAN | List Fns | MiniARC | ACRE | MiniSCAN | List Fns | MiniARC |
| IO | 64.0 | 61.7 | 65.1 | **33.1** | 28.0 | 0.0 | 39.6 | 13.8 |
| SC (N=5) | 65.0 | 61.1 | 65.0 | 31.3 | 29.0 | 0.0 | 38.0 | 13.1 |
| SR (T=3, N=5) | 70.0 | 46.3 | 67.4 | 15.1 | 32.0 | 0.0 | 52.0 | 9.2 |
| T=1, N=1 | 78.2 | 77.0 | 51.6 | 5.9 | 45.0 | 46.0 | 42.4 | 3.8 |
| T=1, N=5 | 79.8 | 86.6 | 62.4 | 12.8 | 48.0 | 70.0 | 52.4 | 9.2 |
| T=3, N=1 | 77.8 | **98.2** | 61.7 | 10.1 | 47.0 | **95.0** | 52.8 | 6.9 |
| T=3, N=5 | **82.5** | 93.3 | **71.2** | 18.7 | **59.0** | 85.0 | **61.2** | **14.6** |

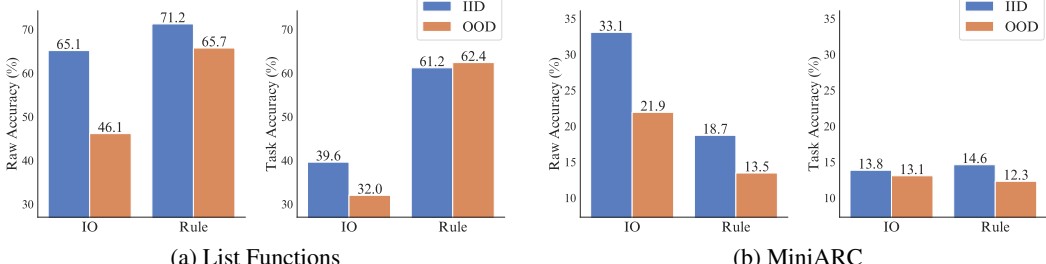

            (a) List Functions                         (b) MiniARC

Figure 2: Results for IID and OOD examples. For OOD evaluations, we sample longer lists for List Functions and annotate larger grids for MiniARC. IO prompting generally experiences more significant performance degradation compared to rule prompting (i.e., iterative hypothesis refinement).

formance, leading to the best accuracy on the majority of datasets.[8] It is important to emphasize that both iterative refinement and external feedback are essential. Simply sampling more predictions and taking a majority vote (as SC prompting), does not necessarily improve performance. This might due to the fact that increasing the temperature for sampling results in many incorrect predictions. In contrast, increasing the number of hypotheses performs better due to the hypotheses selection process. An iterative approach that uses the LM itself as an interpreter (i.e., SR) is also insufficient. We observe performance substantially degrades when replacing the symbolic interpreter with an LM, suggesting that the LM can excel as a hypothesis proposer but performs poorly as an interpreter.

We also observe a significant discrepancy between raw accuracy and task accuracy, especially for IO prompting and SC prompting. Since these evaluations directly predict output for each individual example, there is no guarantee that the LM is solving the task following the underlying rules. In fact, the mismatch between raw accuracy and task accuracy indicates some correct predictions might be generated without using the expected computation. In contrast, rule prompting (i.e., applying the LM-proposed rules) suffers less from this issue as it re-uses the same rule across all examples.

**OOD Generalization and Interpretability.** In addition to strong performance, iterative hypothesis refinement also enables out-of-distribution (OOD) generalization and improves interpretability of models. For OOD evaluations, we sample longer examples from the ground-truth programs for List Functions,[9] and annotate examples with a larger grid for MiniARC. We evaluate performance on these OOD examples while fixing the seen examples. We show the results in Figure 2. We observe a significant performance drop for OOD examples when using IO prompting. However, the degradation is less severe for rule prompting except task accuracy on MiniARC, suggesting the LM likely solves the task using generalizable operations. While IO prompting still achieves better raw accuracy on MiniARC in OOD evaluations, the performance gap between it and rule prompting is reduced. Rule prompting also allows us to examine the intermediate operations, thus improving the interpretability of models. We show examples of LM-induced rules in Table 2 and Table 12.

---

[8]We observe T=3, N=1 performs better than T=3, N=5 on MiniSCAN, potentially because the dataset is designed to evaluate compositional generalization, and thus high accuracy over seen examples does not necessarily translate to high accuracy on unseen examples. Since iterative refinement only uses accuracy over exemplars as the scoring function, it might overfit to exemplars and select hypotheses that are less generalizable.

[9]Although we can theoretically sample longer sequences for MiniSCAN, we did not consider that setup, as the limited exemplars could lead to underspecification and result in multiple plausible parses due to ambiguity.

## 4 LMs ARE PUZZLING INDUCTIVE **REASONERS**

Despite the strong performance of iterative refinement on inductive reasoning, we identify several puzzling behaviors of LMs that seems to differ from human intuition (Fukushima, 1986; Heit, 2000). We include related human studies and human evaluations in Appendix C.[10]

### 4.1 LMs STRUGGLE WITH APPLYING THEIR PROPOSED RULES

Previous results demonstrate that LMs perform as effective hypothesis proposers but poor interpreters. Here we examine the extent to which LMs "understand" the rules they propose. Specifically, given the rules induced from previous experiments, we test whether LMs can apply these rules to novel examples. We should expect comparable performance if LMs understand their own proposed rules. Results are shown in Figure 3. We observe a consistent performance drop when using the LM interpreter as opposed to the symbolic interpreter. This issue is especially significant on datasets like MiniSCAN, where rule application involves complex and recursive operations.

This performance inconsistency between rule induction and rule application reveals a counter-intuitive behavior of LMs on inductive reasoning. Intuitively, once humans have induced a rule, they can use this rule in novel scenarios. However, LMs struggle with applying the rule, even if the rule was derived from themselves. Note that prior work has provided evidence suggesting that LMs might fall short on solving symbolic tasks (Dziri et al., 2023), and we do not claim that we should expect using an LM as the interpreter perform as effectively as a symbolic interpreter. However, the gaps are often so large (e.g., task accuracy dropping from more than 80% to almost-zero in MiniSCAN) that they are still nonetheless strong indicators of LMs' puzzling behaviors.[11] In particular, LMs are able to generate meaningful hypotheses and improve them iteratively, but simultaneously fail to understand their proposed rules. This observation can be loosely related to other inconsistencies observed between generation and recognition in existing LMs (West et al., 2023). For instance, while LMs can identify errors in their own generations (Agrawal et al., 2023; Zhang et al., 2023b), they may also fail to validate a correct answer generated by themselves (Li et al., 2023).

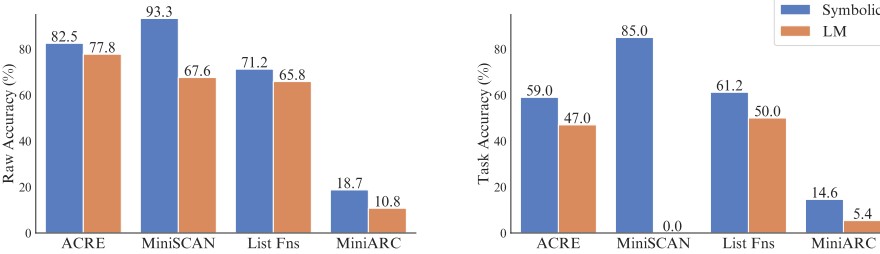

Figure 3: Raw accuracy (left) and task accuracy (right) when applying the LM's proposed rules using symbolic interpreters or the LM itself as the interpreter.

### 4.2 LMs ARE BRITTLE TO EXAMPLE PERTURBATIONS

Our experiments so far only consider well-formed input-output pairs: we assume there always exists at least one ground-truth $f$ such that applying $f$ to the inputs will yield the corresponding outputs. We also assume examples are presented in the format that close to LM's pre-training distribution. However, in practice, low-quality examples are ubiquitous. Humans can often reason robustly despite a certain level of noise, such as disregarding typos or a few erroneous examples (Fukushima, 1986; Heit, 2000). We now investigate if LMs can similarly make robust inductions. We use iterative hypothesis refinement with $T = 3$ and $N = 5$, which has the strongest performance in our main experiments. We include results using other models and configurations in Appendix B.3.

**Noisiness of Exemplars.** We first study LMs' robustness to noisy examples. Specifically, we use List Functions and introduce noise into a certain percentage of exemplars by randomly replacing 1-2

---

[10]While we tried to provide a head-to-head comparison between LMs and humans, our human studies did not cover all experiments conducted with LMs. Therefore, we cannot assert how human participants would perform in certain setups. We leave evaluating human performance more comprehensively as future work.

[11]More advanced prompting techniques, such as SC prompting and zero-shot chain-of-thought prompting, also do not bridge the gap (see Appendix B.3 for details).

elements with other numbers in the outputs. We perturb 12.5%, 25%, and 50% of the examples, out of a total of 8 exemplars. We show results in Figure 4a. We find the LM performs substantially worse even with a single noisy example, and its performance consistently decreases as the amount of noise increases. Although explicitly instructing the LM to consider noisy examples (dashed line) mitigates this issue, the performance degradation remains significant (see Table 17 for the exact instructions). This brittleness raises another concerns about their otherwise promising performance.[12]

**Familiarity of Exemplars.**   Next we study if LMs are robust to example representation. We examine this by varying the familiarity of exemplars, i.e., how well the examples are represented in the LMs' pre-training data. As rules represent higher-level abstraction, ideally we should expect LMs' performance to be independent of their specific instantiations (Newell, 1980). We use MiniSCAN dataset and re-generate new examples using the same grammar rules but with varied output vocabularies. We consider two variants: the first involves pseudowords as inputs with abstract English concepts as outputs (e.g., dax $\rightarrow$ RED), as the original setup in Lake et al. (2019). The second uses pseudowords for both inputs and outputs (e.g., dax $\rightarrow$ zup). The results are shown in Figure 4b. LMs' performance drops when the output representation deviates from their pre-training distribution. In such case, even an iterative approach cannot compensate for this degradation.[13]

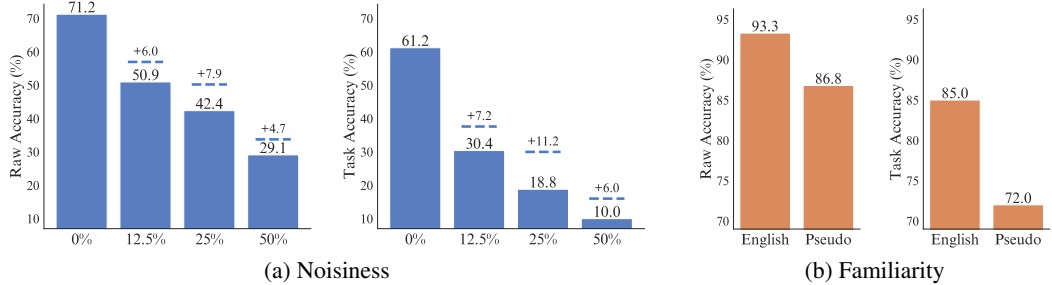

|            (a) Noisiness            |            (b) Familiarity            |

Figure 4: (a) Varying example noisiness by perturbing a certain percentage of exemplars on List Functions. Dashed lines refer to results where we explicitly instruct LMs to consider noisy examples. (b) Varying example familiarity by using English words or pseudo-words as outputs on MiniSCAN.

## 4.3   LM-INDUCED RULES VS. HUMAN-INDUCED RULES

We have provided empirical evidence suggesting some discrepancies between inductive reasoning of LMs and humans. We now qualitatively examine if LM-induced rules are distinguishable from human-induced rules. We conduct analysis on List Functions and MiniARC, as they contain various concepts and represent tasks where the LM succeeds and fails, respectively. We randomly sample 50 tasks per dataset and conduct similar human studies by asking crowdworkers to write the rules.

We show example LM-induced rules and human-induced rules in Table 2. For List Functions where the LM achieves strong performance, the LM can often induce rules that are comparable to or even better than those from humans, with some exceptions where it incorrectly explains the pattern. On MiniARC, however, it tends to generate rules that are difficult to interpret, often involving verbose and complex descriptions. In contrast, similar to Acquaviva et al. (2022), we find that humans often use pragmatic communication strategies that go beyond pattern descriptions. For instance, they frequently draw upon physical commonsense knowledge (e.g., "drop or lift an object", "fill in each box"), use high-level actions (e.g., "copy or extend the block", "mirror the group"), and connect to real-world concepts (e.g., "staircase", "Tetris"). They also pose questions (e.g., "which color is more common in a row and by how many?") and utilize algebraic expressions (e.g., "if a number is repeated $n$ times then only output $n-1$ times") to facilitate effective communications.

---

[12]Given the limited number of exemplars, it is possible that our perturbation results in the task being ill-defined. Humans are not necessarily robust to noisy observations either. Therefore, we conduct human study using the same setup and compare the performance between the LM and humans. We find that, while both the LM and humans perform worse on tasks with noisy examples, the relative performance drop of the LM is more significant. See appendix C for details.

[13]Dasgupta et al. (2022) has shown LMs demonstrate human-like content effects on reasoning, i.e., they tend to reason more accurately about situations that align with their existing knowledge and beliefs. Similarly, humans are not perfect abstract reasoners, but they can remain consistent abstract reasoning given sufficient time budget (Evans & Curtis-Holmes, 2005). Our iterative approach attempts to mitigate this issue. While we did observe performance improvement (as compared to the results in Table 7), it does not fully close the gap.

Table 2: Comparison between LM-induced rules and human-induced rules on List Functions (top) and MiniARC (bottom). `0` maps to `black`, `1` maps to `blue`, and `4` maps to `yellow`.

| Examples | LM-induced Rule | Human-induced Rules |
|---|---|---|
| `[97, 97, 97, 97] → [97, 97, 97]` `[4, 4, 4] → [4, 4]` `[33, 0, 4, 1, 2, 24, 66] → []` `[76, 42, 17, 76, 17] → [76, 17]` ... | Remove the last occurrence of each unique number from the input list, but if a number appears more than twice, keep all instances except the last. | **Annotator 1**: Keep the order of the original list but only include integers that are duplicates rom earlier in the list. **Annotator 2**: Output only the repeated numbers. If a number is repeated n times then output only n-1 times. |
|  | If an element in the input array is 4, replace it with 0. If the element is 1 and its left and right neighboring elements are 0, replace it with 1. If the element is 1 and positioned in the last row of the array, replace it with 1. In all other cases, replace the element with 0. | **Annotator 1**: Slide yellow down, if it completes a row, get rid of the row turn the remaining blocks blue with a 1. **Annotator 2**: Drop the object. If a full row is created, delete it, and drop remaining objects. |

We further investigate how LMs refine their hypotheses. While we observe performance improvement over iterations (see Figure 5), we also notice that they tend to make minor modifications, typically by adding exceptions for a specific example, rather than starting from entirely new hypotheses. We observe several cases where the LM adds an "if-else" statement to the rules over iterations. For instance, the LM generates "Remove the value 2 from the input list." in the first iteration and refines it to "Remove the value 6 if it is present in the input list. If not, remove the value 2" in the subsequent iteration. This results in its failure to induce the correct rules if the initial hypothesis is entirely off.

## 5   LIMITATIONS AND DISCUSSIONS

**Tasks.**   Humans perform inductive reasoning in everyday situations (Hume, 1904). However, our experiments mainly focus on synthetic and symbolic tasks, differing from the typical scenarios in which humans perform inductions. We chose our datasets based on two concerns. First, we interact with LMs using prompting. This restricts the number of seen examples due to LMs' limited context lengths. We selected our tasks because they are relatively constrained and well-defined, making it feasible to induce rules from only a few observations.[14] Second, the inaccessibility of the training data complicates the evaluation of LMs' inductive learning abilities. It is challenging to distinguish whether LMs truly induce rules from observed examples or simply recall the fact from their prior knowledge. Therefore, we chose more synthetic and symbolic tasks, as we hypothesize that they are less likely to be present in the pre-training data, thus making inducing rules from observations necessary. Nonetheless, this confounding factor remains unless we fully inspect the training corpus.

**Hyperparameters.**   The goal of this paper is to explore the potentials and limitations of LMs in inductive reasoning, rather than to improve the performance on specific inductive reasoning tasks. Therefore, we did not exhaustively tune hyperparameters ($T$ and $N$) or prompt templates. Our experiments use a maximum iteration $T = 3$ due to the LMs' limited context lengths and a maximum number of hypotheses per iteration $N = 5$. Our results demonstrate the correlations between model performance and these two hyperparameters. We expect improved performance when increasing these two hyperparameters, as suggested by Table 1 and recent work by Wang et al. (2023a).

**Future Directions.**   Our study demonstrates the effectiveness of using LMs as hypothesis proposers. We show that, when paired with symbolic interpreters, LMs can achieve strong performance

---

[14]The question of how many examples are required for valid induction remains a research question (Osherson et al., 1990; Heit, 2000). Arguably, one might only obtain partial observation, and there are cases where humans perform one-shot induction. However, determining the minimum number of examples necessary for induction is outside the scope of this paper. We believe our tasks suit our evaluation purposes despite their simplicities.

through iterative hypothesis refinement. However, out-of-the-box LMs struggle to solve inductive tasks on their own. This strengthens the need to explore neuro-symbolic approaches to utilize the strengths of both components (Ni et al., 2023; Wong et al., 2023, *i.a.*). Our study also only focuses on a fixed set of exemplars. Future work could explore methods to dynamically select the best exemplars. Additionally, our analyses identify several counter-intuitive model behaviors, highlighting the importance of understanding model behaviors and improving their robustness as future work.

## 6 RELATED WORK

**Inductive Reasoning with LMs.**  Existing studies on inductive reasoning capabilities of pre-trained large LMs (Gendron et al., 2023; Yang et al., 2022; Moskvichev et al., 2023; Mirchandani et al., 2023; Tang et al., 2023; Xu et al., 2023a; Han et al., 2023; Xu et al., 2023b; Alet et al., 2021; Webb et al., 2023) primarily use IO prompting. They focus on evaluating the accuracy of unseen examples but often overlook any intermediate operations. As we argue in our study, this evaluation lacks interpretability and can conflate with LMs' rule application abilities. We instead investigate an alternative evaluation by inducing rules from LMs. Similarly, Honovich et al. (2023) uses LMs to induce instructions from examples, but it only studies dataset-level instructions for simple tasks. Concurrent work (Wang et al., 2023a) that proposes hypothesis search is closest to ours, but we focus on understanding the potentials and limitations of LMs rather than improving LMs' performance.

**Language Hypotheses Optimization.**  Many studies have explored the optimization of hypotheses over the space of natural language. Prior work on latent language for concept learning has mostly focused on few-shot image classification, and often involves training models (Andreas et al., 2018; Mu et al., 2020). Vong & Lake (2022) uses a pre-trained LM, but does not involve refining hypotheses iteratively. Some studies adopt similar iterative frameworks but focus on describing differences between text distributions (Zhong et al., 2022; 2023) or data patterns (Singh et al., 2022). While these hypotheses are relatively coarse-grained, our tasks require fine-grained hypotheses with high precision. Our study shows that, in such cases, a symbolic interpreter is essential to ensure the quality of hypotheses. Additionally, the iterative refinement approach is also related to a line of work on iterative prompting with execution feedback for synthesizing programs (Chen et al., 2023a; Olausson et al., 2023; Haluptzok et al., 2022; Key et al., 2022; Jiang et al., 2023; Zhang et al., 2023a). However, most of these studies use natural language descriptions, sometimes supplemented with optional examples, while ours only use input-output specifications.

**Bayesian Concept Learning.**  Classical approaches to induction primarily follow a Bayesian paradigm: they start with a hypothesis space, compute the posterior distribution using Bayes' Rule, and update beliefs based on observations (Tenenbaum et al., 2011; Lake et al., 2015; Xu & Tenenbaum, 2007; Tenenbaum, 1999; Thaker et al., 2017; Kemp & Tenenbaum, 2009). The main challenge of these methods is the trade-off between expressiveness of hypothesis space and computational cost of posterior inference. Therefore, many of them resolve on searching over a constrained rule-based hypothesis space, such as probabilistic programs (Nosofsky et al., 1994; Piantadosi et al., 2016; Goodman et al., 2008; Bramley et al., 2018; Ellis et al., 2022; 2023). However, a domain-specific formulation of Language of Thought (Fodor, 1975; Erdogan et al., 2015; Saad et al., 2019; Tian et al., 2020; Sablé-Meyer et al., 2022) is often limited. Ellis (2023) addresses this by performing Bayesian inference over natural language. Our approach shares similar spirits of Bayesian models, but instead leverages LMs to generate and refine hypotheses via iterative prompting.

## 7 CONCLUSION

In this paper, we study the inductive reasoning capabilities of LMs and how their inductive reasoning behaviors differ from those of humans. We conduct this investigation through iterative hypothesis refinement, an approach that closely mirrors human inductive process. Iterative refinement operates as a three-step process: hypotheses generation, selection, and refinement. Through our experiments, we find that LMs excel as hypothesis proposers, achieving strong performance on most datasets when coupled with symbolic interpreters. However, we also identify several counter-intuitive behaviors, suggesting that LMs simultaneously behave as puzzling inductive reasoners. For instance, they struggle with applying their own proposed rules and are brittle to minor perturbations. Our study reveals the paradoxical inductive capabilities of LMs and sheds light on both the potentials and limitations of LMs in inductive reasoning tasks.

ACKNOWLEDGMENTS

We thank Jiangjie Chen, Peter Hase, Aniruddha Nrusimha, Kyle Richardson, Zhaofeng Wu, and AI2 Mosaic team for helpful comments and discussions. We thank Jena Hwang, Yufei Tian, and Huirong Wen for the help with human study and data annotation. This work was supported in-part by DARPA MCS program through NIWC Pacific (N66001-19-2-4031) and NSF (DMS-2134012). LQ, BW, and YK were partially supported by MIT-IBM Watson AI and an Amazon award.

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

## A  DATASET DETAILS

We show the dataset statistics in Table 3 and include the full dataset details below.

**ACRE**  Following Gendron et al. (2023), we use textual representations of the original images by representing each object with its corresponding natural language description. We also experimented with a symbolic representation in which each object is represented as an integer, but observed similar performance. We sampled 100 tasks from the original dataset for our experiments.

**MiniSCAN**  We use examples from Lake et al. (2019), but randomly sample pseudowords for the

Table 3: The number of tasks per dataset, the numbers of seen examples per task, and unseen examples per task.

| Dataset | # Tasks | # Seen | # Unseen |
|---|---|---|---|
| ACRE | 100 | 6 | 4 |
| MiniSCAN | 100 | 14 | 10 |
| List Functions | 250 | 8 | 8 |
| MiniARC | 130 | 3 | 3 |

inputs. We did not consider English words because of potential issues of data contamination (Dodge et al., 2021; Magar & Schwartz, 2022, *i.a.*) and uncontrolled lexical exposure (Kim et al., 2022a). The outputs use color names following Akyurek & Andreas (2021); Nye et al. (2020); Patel et al. (2022). We generated a total of 100 tasks for our experiments.

**List Functions**  We use the original dataset (Rule, 2020), which consists of a total of 250 tasks. Due to the limited context lengths of LMs, we only use the first 16 examples from BIG-Bench (bench authors, 2023): 8 for seen examples and 8 for unseen examples. We manually examined the exemplars and found 8 examples are generally sufficient to describe the pattern. Our preliminary experiments also indicated that adding more examples did not improve performance.

**MiniARC**  We use the data from Kim et al. (2022b). Although the original release contains 149 tasks, we heuristically filter out tasks that require heavy pattern matching, such as mapping one specific shape to another. Such tasks are typically difficult to describe in natural language at an abstract level. Therefore, we did not consider them for our evaluations. As we only evaluate text-only models, we use textual representations of the original visual grids by mapping each cell to a corresponding integer (Gendron et al., 2023; Moskvichev et al., 2023).

## B  ADDITIONAL RESULTS

### B.1  OTHER LANGUAGE MODELS

We use GPT-4 for the main experiments, but our observations remain consistent across other LMs, as shown in Table 4. We evaluate GPT-4, GPT-3.5, Claude-2, and LLaMA2-70B using IO prompting and iterative hypothesis refinement, as they are best representatives of two different evaluations. For GPT-3.5 and Claude-2, we observe similar trends except both models underperform GPT-4. Rule prompting achieves higher accuracy than IO prompting on ACRE and MiniSCAN and shows better consistency between raw accuracy and task accuracy. However, these models sometimes lag behind the baseline on tasks involving complex rules, such as List Functions and MiniARC. For LLaMA2-70B, we only observe improvement using rule prompting on ACRE. For tasks where we constrain hypothesis representations, some models' rules appear ill-formed. Many responses from GPT-3.5 and LLaMA2-70B are also truncated due to their limited context length. This suggests

that iterative hypothesis refinement is most effective when coupled with an LM that is capable of proposing meaningful hypotheses and tracking long context.

Table 4: Results on IO prompting and rule prompting (i.e., hypothesis refinement) using different models. We use T=3, N=5 configuration for iterative hypothesis refinement.

| Model | Method | Raw Accuracy | | | | Task Accuracy | | | |
|---|---|---|---|---|---|---|---|---|---|
| | | ACRE | MiniSCAN | List Fns | MiniARC | ACRE | MiniSCAN | List Fns | MiniARC |
| GPT-4 | IO | 64.0 | 61.7 | 65.1 | **33.1** | 28.0 | 0.0 | 39.6 | 13.8 |
| | Rule | **82.5** | **93.3** | **71.2** | 18.7 | **59.0** | **85.0** | **61.2** | **14.6** |
| GPT-3.5 | IO | 56.2 | 14.3 | **55.1** | **18.6** | 12.0 | 0.0 | 27.6 | **8.5** |
| | Rule | **71.5** | **29.3** | 42.2 | 4.6 | **44.0** | **8.0** | **35.6** | 3.8 |
| Claude-2 | IO | 51.7 | 23.4 | **51.4** | **24.7** | 6.0 | 0.0 | 24.4 | 10.8 |
| | Rule | **79.2** | **41.2** | 42.8 | 7.2 | **55.0** | **13.0** | **36.0** | 6.2 |
| LLaMA2-70B | IO | 51.7 | **6.8** | **30.5** | **9.0** | 10.0 | 0.0 | **8.4** | **1.5** |
| | Rule | **64.5** | 0.0 | 9.2 | 2.1 | **29.0** | 0.0 | 6.0 | 0.8 |

Similar to experiments in §4, we show the comparisons between symbolic interpreters and LMs as interpreters for rule application using other models in Table 5. We show results on varying example distribution using different models and configurations in Table 6 and Table 7. All results remain consistent with the findings in the main experiments.

Table 5: Results on applying the LM's proposed rules using symbolic interpreters or the LM itself as the interpreter using different models.

| Model | Interpreter | Raw Accuracy | | | | Task Accuracy | | | |
|---|---|---|---|---|---|---|---|---|---|
| | | ACRE | MiniSCAN | List Fns | MiniARC | ACRE | MiniSCAN | List Fns | MiniARC |
| GPT-3.5 | Symbolic | 71.5 | 29.3 | 42.2 | 4.6 | 44.0 | 8.0 | 35.6 | 3.8 |
| | LM | 65.0 | 3.0 | 36.8 | 3.1 | 25.0 | 0.0 | 24.0 | 1.5 |
| Claude-2 | Symbolic | 79.2 | 41.2 | 42.8 | 7.2 | 55.0 | 13.0 | 36.0 | 6.2 |
| | LM | 75.8 | 4.0 | 36.0 | 3.1 | 43.0 | 0.0 | 24.8 | 0.0 |
| LLaMA2-70B | Symbolic | 64.5 | 0.0 | 9.2 | 2.1 | 29.0 | 0.0 | 6.0 | 0.8 |
| | LM | 59.2 | 0.0 | 7.1 | 1.7 | 12.0 | 0.0 | 2.0 | 0.0 |

Table 6: Results on varying example noisiness using different models and configurations. We introduce noise by perturbing a certain percentage of exemplars on List Functions.

| Model | Configuration | Raw Accuracy | | | | Task Accuracy | | | |
|---|---|---|---|---|---|---|---|---|---|
| | | 0% | 12.5% | 25% | 50% | 0% | 12.5% | 25% | 50% |
| GPT-4 | T=1, N=1 | 51.6 | 49.6 | 36.7 | 23.1 | 42.4 | 38.8 | 23.6 | 12.4 |
| GPT-3.5 | T=3, N=5 | 42.2 | 27.0 | 23.2 | 20.6 | 35.6 | 15.6 | 12.8 | 12.0 |
| Claude-2 | T=3, N=5 | 42.8 | 25.9 | 19.8 | 15.4 | 36.0 | 13.2 | 8.0 | 4.4 |

Table 7: Results on varying example familiarity using different models and configurations. We use English words or pseudo-words as outputs on MiniSCAN.

| Model | Configuration | Raw Accuracy | | Task Accuracy | |
|---|---|---|---|---|---|
| | | English | Pseudo | English | Pseudo |
| GPT-4 | T=1, N=1 | 77.0 | 72.0 | 46.0 | 38.0 |
| GPT-3.5 | T=3, N=5 | 29.3 | 19.5 | 8.0 | 3.0 |
| Claude-2 | T=3, N=5 | 41.2 | 41.0 | 13.0 | 9.0 |

## B.2 MULTIMODAL MODEL

Since the MiniARC dataset requires visual understanding, evaluating text-only models using textual representations may not be optimal. Therefore, we also evaluate the performance of the multimodal model that allows visual inputs. We use GPT-4V (`gpt-4-vision-preview`), which was released in November 2023, for our experiments. We consider two representations of the visual grids:

Table 8: Results on MiniARC using GPT-4V. We show the results of IO prompting and rule prompting, as well as the results when applying the model's proposed rules using the symbolic interpreter or GPT-4V as the interpreter. We use a T=3, N=5 configuration for iterative hypothesis refinement.

| Method | Interpreter | Raw Accuracy | | Task Accuracy | |
| --- | --- | --- | --- | --- | --- |
| | | Color | Number | Color | Number |
| IO | - | 1.3 | 12.2 | 0.0 | 3.8 |
| Rule | Symbolic | 6.0 | 9.7 | 4.6 | 8.5 |
| Rule | GPT-4V | 0.5 | 3.1 | 0.0 | 0.8 |

color and number. We use an individual image for each input and output (see Table 18 for prompts and examples). For iterative hypothesis refinement, we use GPT-4 to translate hypotheses due to the rate limit of GPT-4V. For IO prompting and rule application, we ask the model to generate the textual representation of the visual grid, representing each cell as an integer.[15] We show results in Table 8. The performance of GPT-4V is significantly worse than that of GPT-4, which is consistent with the results in Mitchell et al. (2023). Performance with color representation lags behind when compared to numerical representation. Similarly, we find that using GPT-4V as a rule interpreter consistently underperforms using the symbolic interpreter.

## B.3 ABLATIONS

**Prompting Techniques for Rule Application.** We only use standard prompting for rule application in §4.1. Here, we study whether more advanced prompting techniques improve LMs' rule application performance. We consider two alternatives: self-consistency prompting (SC; Wang et al., 2023b) and zero-shot chain-of-thought prompting (0-CoT; Kojima et al., 2022; Nye et al., 2021; Wei et al., 2022). Similar to our main experiments, SC selects the most consistent output from multiple responses by taking a majority vote. Following Kojima et al. (2022), 0-CoT adds "Let's think step by step." at the end to encourage LMs to reason. We show results in Table 9. We do not observe significant performance differences across these methods, except on ACRE, where 0-CoT underperforms other methods in task accuracy. This could potentially be attributed to the possibility that LMs do not truly understand their own proposed rules; therefore, encouraging reasoning might result in worse performance.

Table 9: Results on using LMs as interpreters for rule application with different prompting techniques. We compare standard prompting, zero-shot chain-of-thought (0-CoT) that adds "Let's think step by step" at the end, and self-consistency (SC) that selects the output by taking a majority vote.

| Method | Raw Accuracy | | | | Task Accuracy | | | |
| --- | --- | --- | --- | --- | --- | --- | --- | --- |
| | ACRE | MiniSCAN | List Fns | MiniARC | ACRE | MiniSCAN | List Fns | MiniARC |
| Standard | 77.8 | 67.6 | 65.8 | 10.8 | 47.0 | 0.0 | 50.0 | 5.4 |
| 0-CoT | 73.2 | 65.5 | 61.2 | 12.1 | 25.0 | 0.0 | 48.4 | 6.9 |
| SC (N=5) | 77.0 | 67.5 | 66.3 | 9.7 | 46.0 | 0.0 | 50.8 | 4.6 |

**Representation of Hypothesis.** We investigate how the representation of hypothesis affects rule induction. We use programming language hypotheses for List Functions and MiniARC. We consider this alternative as existing studies have shown that prompting LMs to generate programs improves the model's performance on complex reasoning task (Gao et al., 2023; Chen et al., 2022; Hu et al., 2023, *i.a.*). Directly using programming language hypotheses also eliminates the potential issue of mistranslation between natural language and code. As shown in Table 10, both programming language and natural language hypotheses achieve comparable performance, suggesting programming language can be a powerful alternative for these tasks.

---

[15]We also experimented with using a single image for all exemplars and using an image for each input-output pair, but found that neither approach achieved good results, similar to findings in Mitchell et al. (2023). Our preliminary experiments also suggested that representing each cell as a color string performs worse than representing each cell as an integer.

Table 10: Results on using alternative hypothesis representation. We compare natural language hypotheses (NL) and programming hypotheses (Python) on List Functions and MiniARC.

|  | **Raw Accuracy** | | **Task Accuracy** | |
| --- | --- | --- | --- | --- |
| Hypothesis | List Fns | MiniARC | List Fns | MiniARC |
| NL | 71.2 | 18.7 | 61.2 | 14.6 |
| Python | 72.5 | 18.1 | 65.6 | 13.8 |

**Task-specific Heuristics.**  One reason why humans can learn new concepts from limited examples is their strong inductive biases or prior knowledge (Lake et al., 2017). We evaluate whether imposing task-specific heuristics influences LMs' inductive reasoning behaviors. Specifically, we use the MiniARC dataset, which involves visual understanding, and thus object-related heuristics could potentially be beneficial. Similar to Wang et al. (2023a), we provide explicit task-specific heuristics[16] in the prompt for hypothesis generation, as shown in Table 11. We observe that the LM-induced rules become more human-readable. The LM starts to use visual concepts (e.g., "square", "rectangle", "L-shape", "U-shape") and common transformations (e.g., "reflection", "mirror", "rotate the grid 90 degrees clockwise"). We show example LM-induced rules in Table 12. However, this behavior appears only in a fraction of examples, and the rules induced by LMs are still generally distinguishable from those induced by humans. It is possible that incorporating additional guidance or adding human-induced rules as few-shot examples could encourage LMs to use pragmatic communication strategies. We leave exploring these alternatives as future work.

Importantly, imposing task-specific heuristics does not necessarily improve performance. Iterative hypothesis refinement with $T = 3$ and $N = 5$ achieves a raw accuracy of 17.8% and task accuracy of 14.6%, comparable to results without task-specific heuristics. One possible reason is the integer-color mapping introducing additional overhead, as LMs frequently refer to both simultaneously in the rule (e.g., "if a pixel is green (3), then change it to red (2)"). This could also potentially be explained by observations in Acquaviva et al. (2022): human communication is expressive yet ambiguous. Therefore, the more human-readable rules might require extra efforts to ensure precision and improve performance.

Table 11: Task-specific hypothesis generation prompt for MiniARC.

```
Generate a rule that maps the following inputs to their corresponding outputs.
Both the input and output are 5x5 grids of integers, with each integer
representing a colored pixel in the visual grid.  The integers can be mapped to
colors as follows:

0:  Black; 1:  Blue; 2:  Red; 3:  Green; 4:  Yellow; 5:  Grey; 6:  Fuchsia;
7:  Orange; 8:  Teal; 9:  Brown.

The black cells represent the background.

Hints:  The transformations might include, but are not limited to:

- Movement:  Flipping, rotation, reflection, etc.
- Color:  Swapping and rotating colors between objects, etc.
- Object:  Moving and copying objects, etc.
- Number:  Counting the number of colors, comparing the number of two colors, etc.
- Geometry:  Aligning or completing objects, etc.
- Common sense:  Finding mazes paths, playing Tetris, simulating raindrops, etc.

Please format your rule as follows:

Rule:  <Your rule>
```

---

[16]We provide descriptions of the representative transformations from different categories in  Kim et al. (2022b), as well as the mapping between integers and colors. In most cases, the color of an object does not have a specific meaning. However, in certain categories, such as common sense, specifying the color can provide a useful cue. For instance, in a transformation that simulates raindrops, recognizing an object with blue color as rain could help reasoning.

Table 12: Comparison between LM-induced rules on MiniARC with and without task-specific heuristics. We show examples where heuristics are helpful (top) and not helpful (bottom) for rule induction. 0 maps to black.

| Examples | w/o Heuristics | w/ Heuristics |
|---|---|---|
| | For each row in the input array, the corresponding output row is generated by taking the first element from the current row, the second element from the first row, the third element from the current row, the fourth element from the second row and the fifth element from the first row. If the first element of the current row is 0, then the entire output row becomes 0. | The input grid is mirrored along its main diagonal, i.e., the diagonal from the top-left corner to the bottom-right corner. The color mapping remains the same. |
| | For a given 5x5 matrix, copy the first two values of the first row to the third and fourth positions of the fourth row and the first two values of the second row to the third and fourth positions of the third row. Then, copy the second value of the second row to the last position of the last row. Leave all the other positions unchanged. | The output grid is formed by taking the existing colored cells (non-zero numbers) from the top-left corner of the input grid, and creating a mirror image of this pattern in the bottom-right corner. This mirror image involves not only flipping the pattern along the central vertical and horizontal axes, but also rotating the position of the colors one position to the right in each cell. The rest of the cells remain black (0). |

## B.4 ANALYSIS

Our main experiments demonstrate the effectiveness of the iterative approach. In Figure 5, we show the changes of accuracy over iterations. We observe consistent performance improvements across all datasets, indicating that LMs are capable of refining their hypotheses iteratively. For tasks where LMs achieve strong performance, such as ACRE and MiniSCAN, a limited number of iterations is already sufficient. For tasks like MiniARC, where LMs perform poorly, the trends remain positive after the maximum number of iterations. This suggests potential for further improvements with more iterations when using LMs with longer context lengths.

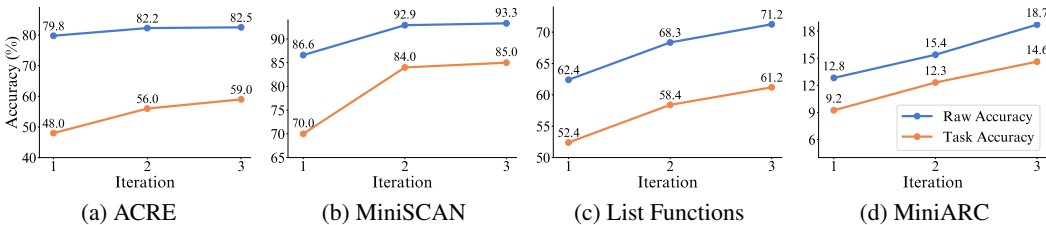

Figure 5: Raw accuracy and task accuracy over iterations.

## B.5 COSTS

We show the average number of API calls and the average cost per task in Table 13. For GPT-4, the cost is computed using $0.03/1K tokens for input and $0.06/1K tokens for output. For GPT-3.5, the cost is computed using $0.0015/1K tokens for input and $0.002/1K tokens for output. Iterative hypothesis refinement, when augmented with a symbolic interpreter, is more cost-efficient than SR,

Table 13: The average number of API calls and the average cost per task.

| Model | Method | # API Calls | | | | Cost (cent) | | | |
|---|---|---|---|---|---|---|---|---|---|
| | | ACRE | MiniSCAN | List Fns | MiniARC | ACRE | MiniSCAN | List Fns | MiniARC |
| GPT-4 | IO | 4.0 | 10.0 | 8.0 | 3.0 | 2.0 | 7.9 | 9.1 | 6.4 |
| | SC (N=5) | 19.8 | 50.0 | 40.0 | 14.8 | 2.0 | 7.9 | 9.1 | 6.4 |
| | SR (T=3, N=5) | 16.5 | 31.6 | 22.2 | 31.0 | 6.5 | 26.7 | 8.1 | 28.1 |
| | T=3, N=5 | 8.2 | 6.3 | 17.4 | 27.2 | 2.3 | 4.5 | 12.0 | 36.5 |
| GPT-3.5 | IO | 4.0 | 10.0 | 8.0 | 3.0 | 0.1 | 0.5 | 0.5 | 0.3 |
| | T=3, N=5 | 9.8 | 11.2 | 21.4 | 27.4 | 0.1 | 0.7 | 0.8 | 1.6 |
| Claude-2 | IO | 4.0 | 10.0 | 8.0 | 3.0 | – | – | – | – |
| | T=3, N=5 | 8.7 | 14.4 | 20.2 | 26.2 | – | – | – | – |

as it reduces the number of API calls required to apply hypotheses. It is also more cost efficient for tasks with a larger number of test examples, e.g., MiniSCAN, as it re-uses the same rule across all examples.

## C  HUMAN STUDIES

### C.1  EXISTING HUMAN PERFORMANCE

Our experiments are largely motivated by cognitive science literature. Here, we collect results from existing human studies to better calibrate the performance of LMs and humans. It is important to note that the exact setups, data, and evaluations in these studies might differ from ours. Therefore, the reported human performance can only be used as a reference but not for direct comparison.

For ACRE, Gopnik et al. (2001) and Sobel et al. (2004) found that 3-4 year-old children are able to identify if an object is a Blicket within 2 trials. For MiniSCAN, Lake et al. (2019) conducted similar human experiments and found humans achieve around 80% average accuracy, with the lowest performance at around 65% and the highest at 88%. For List Functions, Rule (2020) reported the average human performance of 45.9%.[17] For MiniARC, Kim et al. (2022b) did not provide any human experiment results. However, Johnson et al. (2021) evaluated a subset of tasks from ARC (Chollet, 2019) and found that human participants can solve 80% of the tasks, with 65% of tasks being solved by more than 80% of participants. Moskvichev et al. (2023) evaluated human participants on ConceptARC, a variant of ARC, and reported an average human performance of 90.9% in solving test examples. Additionally, Acquaviva et al. (2022) found that human annotators were able to write correct instructions for 88% of the ARC tasks.

### C.2  SETUP

We randomly sample 50 tasks for List Functions and MiniARC and ask crowdworkers to write and evaluate rules. For each task, we ask 3 annotators to write the rule, and for each rule pair, we ask another 3 annotators to evaluate them. For rule evaluation, following prior work (Saha et al., 2022; Chen et al., 2023b), we consider two metrics: clarity and supportiveness. Clarity evaluates whether the rule provides a clear explanation of the transformation from input to output. Supportivenss measures how well the rule align with examples.[18] We use pairwise comparison with 4 labels: LM-induced rule is better, human-induced rule is better, equally good, and equally bad.

We use Amazon Mechanical Turk for all human studies. We select crowdworkers who are located in the US with a HIT approval rate higher 97% and at least 10000 HIT approved. We pay our annotators at a minimal hourly wage of $15. We show the instructions and annotation interfaces for rule induction in Figure 8 and Figure 9, and for rule evaluation in Figure 10 and Figure 11.

---

[17]The human performance was obtained by asking human participants to play a guessing game. It only requires solving unseen examples and did not involve writing rules. See Rule (2020) for details.

[18]Saha et al. (2022) use three metrics for evaluation: clarity, supportiveness, and generalizability. We did not consider generalizability as we directly evaluate on unseen examples. Our pilot experiments also suggest that crowdworkers found it challenging to distinguish between supportiveness and generalizability.

## C.3 RESULTS

**Human Preferences.** We show results of human pairwise comparisons in Figure 6. For List Functions, where the LM achieves high accuracy, LM-induced rules and human-induced rules are comparably clear, but the former are sometimes less supportive. On MiniARC, where the LM performs poorly, we observe a significant performance gap between LM-induced rules and human-induced rules on both clarity and supportiveness. The average inner-annotator agreement, measured by Cohen's Kappa, is 0.13 for clarity and 0.53 for supportiveness.

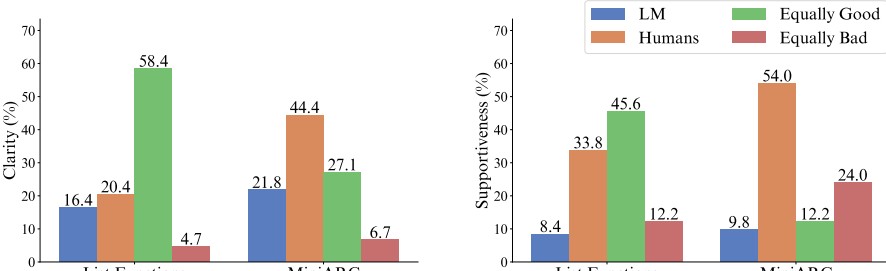

Figure 6: Comparisons of LM-induced rules versus human-induced rules in terms of clarity (left) and supportiveness (right).

**Impacts of Noisy Exemplars.** We conduct similar human experiments using the data with 12.5% noise from §4.2. We consider two setups: one with a hint indicating some examples might be incorrect (comparable to the dashed line in Figure 4a) and one without any hint (comparable to the bar in Figure 4a). We measure the percentage of rules that are preferred or equally good for either the LM or humans, and show the relative performance difference in Table 14. We also show the original human preferences in Figure 7. While both the LM and humans perform worse on tasks with noisy examples, the relative performance drop of the LM is generally more significant. The average inner-annotator agreement, measured by Cohen's Kappa, is 0.15 for clarity and 0.54 for supportiveness.

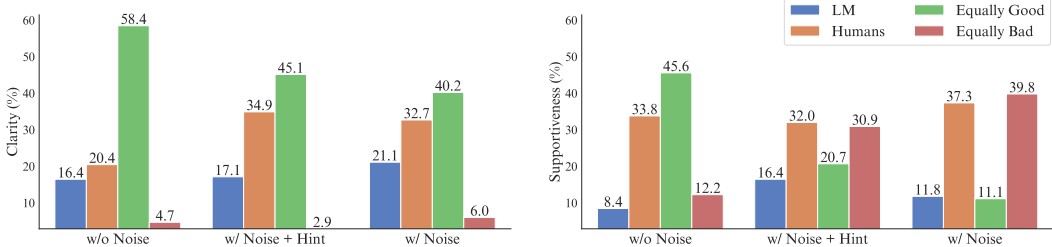

Figure 7: Human preferences for LM-induced rules versus human-induced rules on List Functions, using exemplars without noise, with noise and a hint, and with noise only.

## D PROMPTS AND EXAMPLES

Our experiments use several types of prompts. For rule induction, we query LMs for hypotheses generation and hypotheses refinement. For rule application, we query LMs to apply the rules. We also ask LMs to translate natural language hypotheses to Python programs. We show different types of prompts in Table 15. The values that fill in the placeholders for each dataset along with examples are shown in Table 16, Table 17, and Table 18.

Table 14: Percentage of rules induced by the LM and humans on List Functions that are preferred or equally good, along with the relative difference, when using exemplars without noise, with noise and a hint, and with noise only.

| | **Clarity** | | | | | **Supportiveness** | | | | |
| | | w/ + Hint | | w/ | | | w/ + Hint | | w/ | |
| | w/o | abs. | rel. $\Delta$ | abs. | rel. $\Delta$ | w/o | abs. | rel. $\Delta$ | abs. | rel. $\Delta$ |
|---|---|---|---|---|---|---|---|---|---|---|
| LM | 74.9 | 62.2 | -16.9 | 61.3 | -18.1 | 54.0 | 37.1 | -31.3 | 22.9 | -57.6 |
| Humans | 78.9 | 80.0 | 1.4 | 72.9 | -7.6 | 79.3 | 52.7 | -33.6 | 48.4 | -38.9 |

Table 15: Prompts used in our study. {} refers to a placeholder.

| Type | Prompt |
|---|---|
| Hypothesis Generation | Generate a rule that maps the following inputs to their corresponding outputs.  **{Task description}**

**{Examples}**

Please format your rule as follows:

**{Rule format}** |
| Hypothesis Refinement | Your rule:  **{Rule}**

This rule does not work for the following examples.

**{Feedback}**

Generate a new rule that maps the given inputs to their corresponding outputs.  **{Feedback description}** Please format your rule as follows:

**{Rule format}** |
| Hypothesis Translation | You are an expert Python programmer.  Write a Python function 'fn' for the following rule.  **{Translation Example description}**

Rule:  **{Rule}** |
| Rule Application | Generate an output corresponding to the given input based on the rule.  **{Application Example description}**

Rule:  **{Rule}**

Input:  **{Test input}**
Output: |

Table 16: Prompts and examples for ACRE and MiniSCAN.

| | ACRE | MiniSCAN |
|---|---|---|
| Task Description | Each example is an input-output pair. The input is a list of objects. The presence of certain objects will trigger the light to turn on. The output is either "on" or "off", indicating the state of the light. For each object, determine whether it triggers the light to turn on, does not trigger it, or if it's undetermined. | Your grammar rules should follow the format "<input> -> <output>". Use the prefix "##" to denote a nonterminal symbol. For instance, "##A twice -> ##A ##A". The left-hand side cannot contain repetitive nonterminal symbols; i.e., rules like "##A ##A -> ##A twice" or "##A and ##A -> ##A twice" are not allowed. Ensure that the number of unique nonterminal symbols on the left-hand side matches that on the right-hand side in your rules. For each rule, assign an integer as its priority. A higher priority indicates that the rule should be considered first when generating parses. Try to make your rules as minimal as possible. |
| Application Example Description | Each example is an input-output pair. The input is a list of objects. The presence of certain objects will trigger the light to turn on. The output is either "on", "off", or "undetermined", indicating the state of the light or if the state of the light cannot be determined. The rule indicates whether each object triggers the light to turn on, does not trigger it, or if it's undetermined. | The grammar rules follow the format "<input> -> <output>". The "##" prefix denotes a nonterminal symbol. For instance, ##A twice -> ##A ##A. Each rule has an associated priority. A higher priority indicates that the rule should be considered first when generating parses. The output is a sequence of tokens joined by spaces. |
| Feedback Description | / | / |
| Rule format | Rule: {"object 1": <"on"/"off"/"undetermined">, "object 2": <"on"/"off"/"undetermined">, ...} | Rule 1: <Your rule>
Priority 1: <Your priority>
... |
| Rule | {"blue rubber sphere": "on", "red metal cube": "off"} | Rule 1: siun -> BLUE
Priority 1: 2
Rule 2: #A mcneilt -> #A #A #A
Priority 2: 1 |
| Examples | Input: blue rubber sphere
Output: on
... | Input: siun mcneilt
Output: BLUE BLUE BLUE
... |
| Test input | blue rubber sphere | siun mcneilt |
| Feedback | Input: blue rubber sphere
Expected output: on
Actual output: off
... | Input: siun mcneilt
Expected output: BLUE BLUE BLUE
Actual output: BLUE BLUE
... |

Table 17: Prompts and examples for List Functions and MiniARC. {} is added if we use noisy examples (§4.2).

| | List Functions | MiniARC |
|---|---|---|
| Task Description | / | / |
| Translation Example Description | The input is a list of integers. The output is also a list of integers. | The input is a nested list that represents a 2D grid of integers. The output is also a nested list that represents a 2D grid of integers. |
| Application Example Description | The input is a list of integers. The output is also a list of integers. | The input is a 2D grid of integers. The output is also a 2D grid of integers. |
| Feedback Description | {Note that some examples may be noisy, and you should take this into account when proposing the rule.} | / |
| Rule format | Rule: <Your rule> | Rule: <Your rule> |
| Rule | Remove the first element and the last two elements | For each cell in the input, if the cell value is 1 and all cells in the 3x3 grid surrounding it (including diagonally) are also 1s, then the output value for that cell is 0. If the cell value is 0 and it is surrounded by 1s on all four sides (up, down, left, and right), then the output value for that cell is 7. All other cells in the output should match the corresponding cells in the input. |
| Examples | Input: [9, 7, 1, 8, 2, 3]
Output: [7, 1, 8]
... | Input:
[1, 1, 1, 1, 1]
[1, 0, 0, 0, 1]
[1, 0, 0, 0, 1]
[1, 0, 0, 0, 1]
[1, 1, 1, 1, 1]
Output:
[0, 0, 0, 0, 0]
[0, 7, 7, 7, 0]
[0, 7, 7, 7, 0]
[0, 7, 7, 7, 0]
[0, 0, 0, 0, 0]
... |
| Test input | [3, 8, 2, 5, 4] | [0, 1, 1, 1, 1]
[1, 1, 0, 0, 1]
[1, 0, 0, 0, 1]
[1, 1, 0, 0, 1]
[0, 1, 1, 1, 0] |
| Feedback | Input: [9, 7, 1, 8, 2, 3]
Expected output: [7, 1, 8]
Actual output: [7, 1, 8, 2, 3]
... | Input:
[1, 1, 1, 1, 1]
[1, 0, 0, 0, 1]
[1, 0, 0, 0, 1]
[1, 0, 0, 0, 1]
[1, 1, 1, 1, 1]
Expected output:
[0, 0, 0, 0, 0]
[0, 7, 7, 7, 0]
[0, 7, 7, 7, 0]
[0, 7, 7, 7, 0]
[0, 0, 0, 0, 0]
Actual output:
[1, 1, 1, 1, 1]
[1, 0, 0, 0, 1]
[1, 0, 0, 0, 1]
[1, 0, 0, 0, 1]
[1, 1, 1, 1, 1]
... |

Table 18: Prompts and examples for MiniARC when using multimodal models.

| | Color | Number |
|---|---|---|
| Translation Example Description | The input is a nested list that represents a 2D grid of integers. The output is also a nested list that represents a 2D grid of integers. The integers can be mapped to colors as follows:

0: Black; 1: Blue; 2: Red; 3: Green; 4: Yellow; 5: Grey; 6: Fuchsia; 7: Orange; 8: Teal; 9: Brown. | The input is a nested list that represents a 2D grid of integers. The output is also a nested list that represents a 2D grid of integers. |
| Application Example Description | Represent your output as a 2D grid of integers, using the format below.

[0, 0, 0, 0, 0]
[0, 0, 0, 0, 0]
[0, 0, 0, 0, 0]
[0, 0, 0, 0, 0]
[0, 0, 0, 0, 0]

The integers can be mapped to colors as follows:

0: Black; 1: Blue; 2: Red; 3: Green; 4: Yellow; 5: Grey; 6: Fuchsia; 7: Orange; 8: Teal; 9: Brown. | Represent your output as a 2D grid of integers, using the format below.

[0, 0, 0, 0, 0]
[0, 0, 0, 0, 0]
[0, 0, 0, 0, 0]
[0, 0, 0, 0, 0]
[0, 0, 0, 0, 0] |
| Rule | If a blue square is adjacent (horizontally or vertically) to the central black 2x2 square, change its color to orange. Then, change all other squares to black. | For each cell in the matrix that is '1', change it to '7' if it is neither on the border of the matrix nor adjacent to a '0'. Change all other cells to '0'. |
| Examples | Input:

Output:

... | Input:

Output:

... |
| Test input |  |  |
| Feedback | Input:

Expected output:

Actual output:

... | Input:

Expected output:

Actual output:

... |

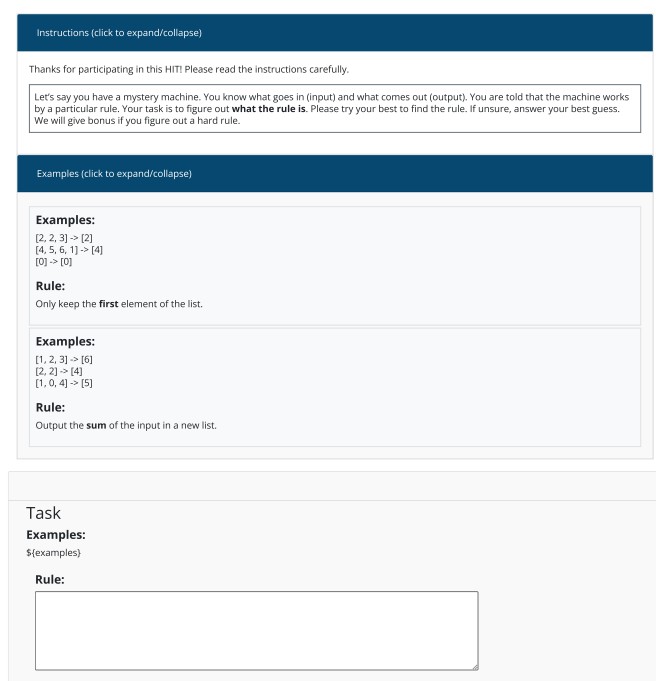

Figure 8: Annotation interface for human rule induction on List Functions.

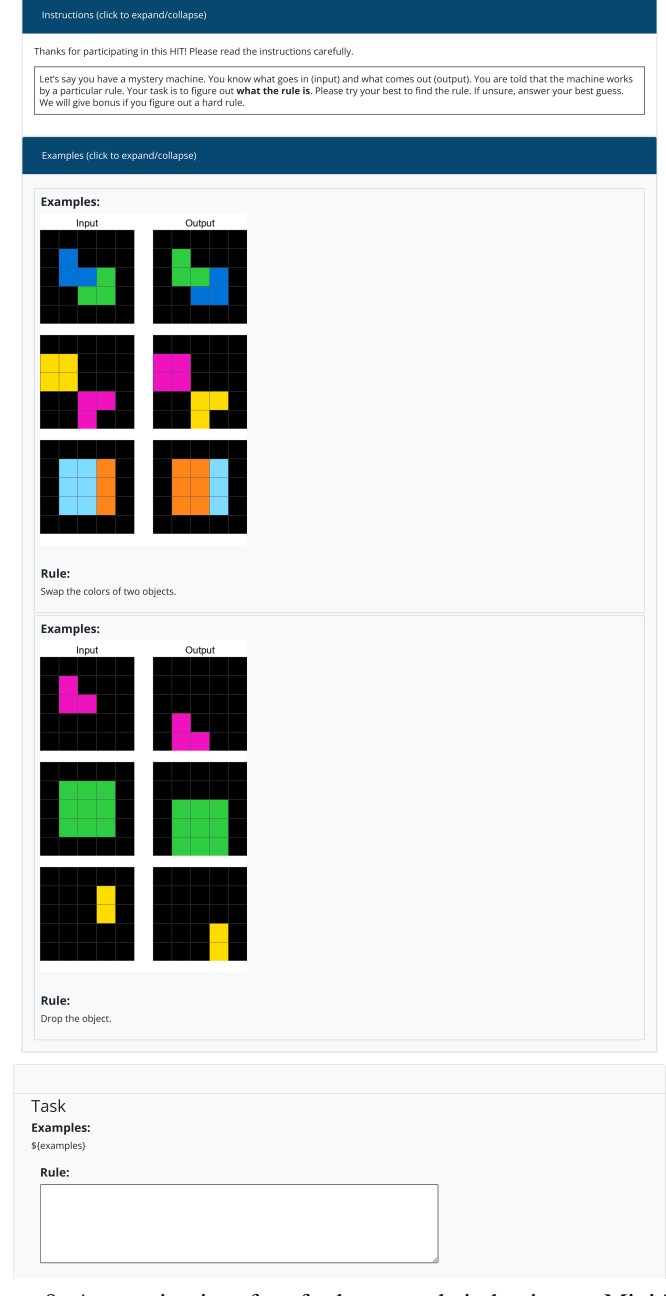

Figure 9: Annotation interface for human rule induction on MiniARC.

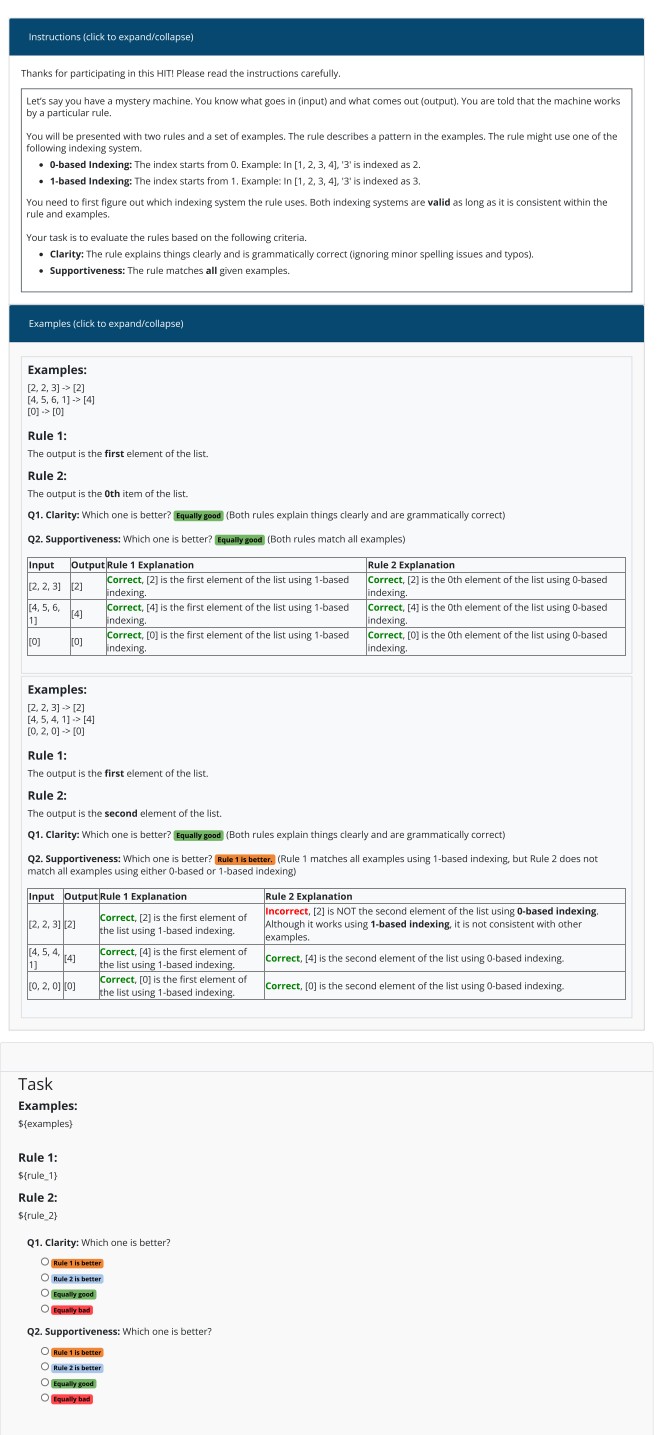

Figure 10: Annotation interface for human rule evaluation on List Functions.

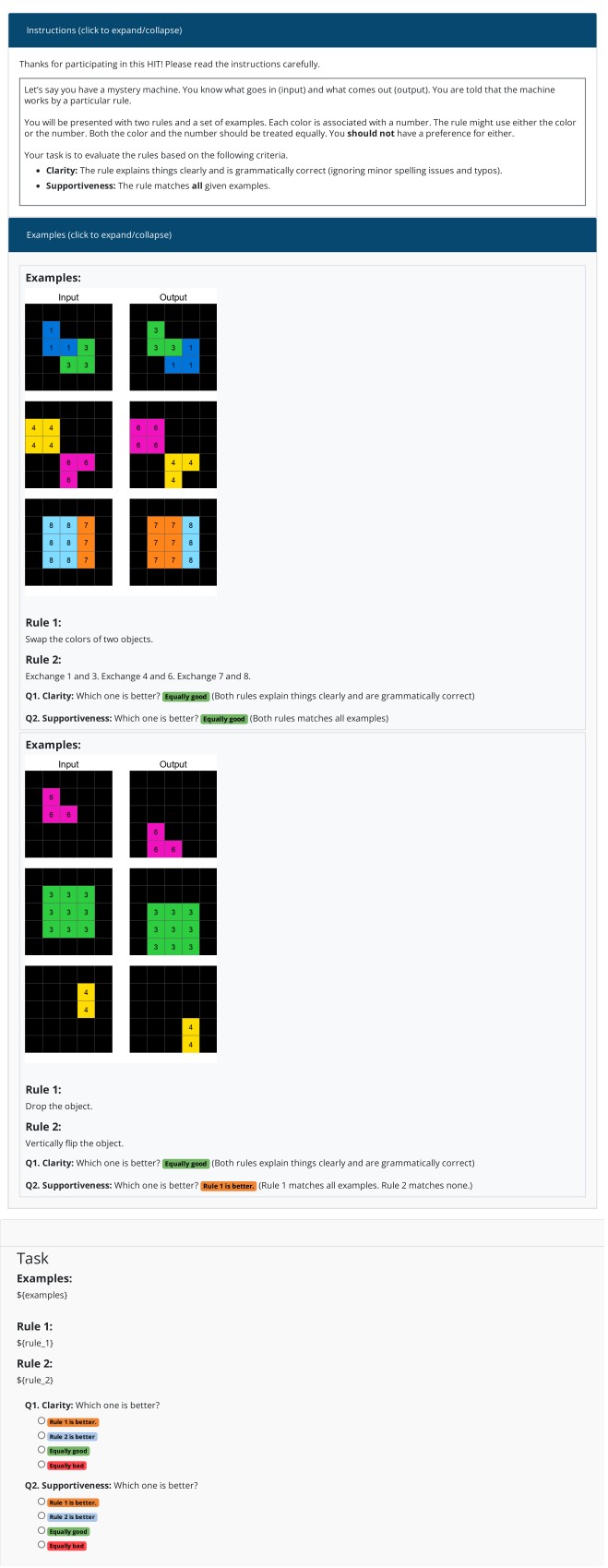

Figure 11: Annotation interface for human rule evaluation on MiniARC.

