# OpenReview forum: "Phenomenal Yet Puzzling: Testing Inductive Reasoning Capabilities of Language Models with Hypothesis Refinement"
_ICLR.cc/2024/Conference — ICLR 2024 oral_

### Official Review · Reviewer_MmPX · 2023-10-22

**Soundness:** 3 good
**Presentation:** 3 good
**Contribution:** 3 good
**Rating:** 8
**Confidence:** 5

**Summary:**

This work examines the inductive reasoning capabilities of LLMs, decomposing this process into two distinct stages: hypothesis proposal and rule application. The results suggest that LLMs are often able to propose reasonable hypotheses, but less reliable at applying those hypothesized rules. A neurosymbolic approach is proposed in which LLM-proposed hypotheses are symbolically implemented, improving performance on multiple inductive reasoning tasks.

**Strengths:**

- Diverse set of tasks
- Thorough evaluation, evaluating range of hyperparameters, metrics, and base models
- The proposed approach, hypothesis refinement, is interesting and yields improved performance across multiple tasks. The approach also has interesting connections to human reasoning.
- The distinction between hypothesis proposal and rule application generates useful insights into the strengths and weaknesses of reasoning in LLMs.

**Weaknesses:**

- The results would be more informative if compared directly with human performance. Do any of these benchmarks contain human performance measures (e.g. I believe that there is already human behavioral data for miniSCAN in the original paper)?
- The results on noisy rule induction are especially difficult to interpret without a human baseline. The authors cite a paper indicating that humans are somewhat robust to noise when inducing rules, but the amount of noise and the specific tasks will matter a lot.
- Throughout the paper, the authors appeal to intuition concerning putative human performance on the benchmarks they consider (e.g. in considering the potential for human reasoners to show a discrepancy between hypothesis generation and rule application), but intuition is not always a reliable guide regarding human performance. It would be good to qualify these statements a bit more (or, to the extent possible, to include a direct comparison with human performance).
- The 'task accuracy' measure seems designed to emphasize the consistency of the symbolic rule application. When looking at raw accuracy, the differences between symbolic and LLM rule application don't look nearly as large. I think this somewhat undermines the claim that their rule application abilities are so much worse than their hypothesis generation abilities.
- It should be emphasized more that miniARC, unlike the other tasks, is a distinctly visual task, in that it requires understanding visual concepts like 'objectness'. It is somewhat unsurprising that a text-only model would show special difficulty on such a task.
- Do the authors have any explanation for why the hypothesis refinement approach achieves worse performance on miniARC relative to the baselines? Given the visual nature of the task, it's not surprising that it doesn't help much, but I was surprised that it actually seems to impair performance.
- The familiarity analysis seems to confound two very different issues: 1) The presence of the same exact problems in the LLM pretraining data (a significant possibility, given the use of pre-existing datasets), and 2) the use of familiar vs. unfamiliar words (e.g. pseudowords). I think it's important to dissociate these concerns. This can be done by creating a new dataset with similar properties, e.g. by replacing the specific pseudowords and colors in miniSCAN (but maintaining the same general pseudoword -> color structure).
- Is the interaction between iid vs. ood and IO vs. hypothesis refinement (figure 2) statistically significant? Also, the text describes the hypothesis refinement results as demonstrating superior robustness to this ood setting, but for miniARC the ood accuracy is actually higher for the IO baseline (even though the difference between iid and ood performance is larger for IO). This should be clarified in the text.
- Were the language model and human hypotheses systematically compared in any way, or only qualitatively inspected? Did the strength of the language model hypotheses correspond to performance on the task? Were there cases where the model performed well on the task despite providing unhuman-like hypotheses?

## Minor comments:
- What setting was used for the 'top p' parameter in GPT-4?
- It would be helpful to either include the variable names in figure 1, or to have a separate figure illustrating the overall flow of the model with the corresponding variable names.
- It would be good to say a bit more about how this work differs from Wang et al (2023). This is concurrent work, so there is no concern about novelty, but it would still be useful to have more discussion of the relationship.
- I found the description of the approach somewhat confusing. Based on the abstract and intro, I was expecting that the hypotheses would be articulated in natural language, and this would somehow be translated into code which is then symbolically executed. It is explained later on (in section 2.2) that the LM also carries out this translation step for list functions and miniARC, but it would be good to provide some hint that this is the case earlier in the section describing the approach. My understanding is that for the other tasks, the LM is prompted so as to ensure hypotheses in a particular format, which can then be automatically parsed, is that correct? It would be helpful to clarify this (in section 2 it says that the hypotheses are 'constrained', but it's not immediately clear what that means).
- I found this sentence, 'However, they also behave as puzzling inductive reasoners, showing notable performance gaps in rule induction (i.e., identifying plausible rules)…' to be confusing, because it seems to say that LLMs are bad at proposing rules, even though it was just stated that they are good at this. This also seems misaligned with the results. It seems like this sentence should instead emphasize the rule application specifically (though, as mentioned above, it's not clear how significant this discrepancy really is).
- The authors might consider citing this related work on analogical reasoning (a special case of inductive reasoning) in LLMs: Webb, T., Holyoak, K. J., & Lu, H. (2023). Emergent analogical reasoning in large language models. Nature Human Behaviour.

**Questions:**

I have listed some questions in the previous section. I would be happy to raise my score if some of these issues can be addressed.

---

> ### Author Response · Authors · 2023-11-19
> **Response to Reviewer MmPX (1/n)**
>
> Thank you so much for your insightful questions and constructive feedback!
> ### Response to weakness
>
> >  The results would be more informative if compared directly with human performance. Do any of these benchmarks contain human performance measures (e.g. I believe that there is already human behavioral data for miniSCAN in the original paper)?
>
> Thanks for the suggestion! All of our experiments are largely motivated by cognitive scene literature. Therefore, they do have existing human performance as a reference. Note the exact setups, data, and evaluations in these studies might differ from ours. We include all existing human studies results in Appendix C.1 and refer it when discussing main results in Section 3.
>
> >  The results on noisy rule induction are especially difficult to interpret without a human baseline. The authors cite a paper indicating that humans are somewhat robust to noise when inducing rules, but the amount of noise and the specific tasks will matter a lot.
>
> We conducted human study using the same setup and added results in Appendix C.3. We observe that both the LM and humans perform worse on tasks with noisy examples. However, the relative performance drop of the LM is more significant. We clarified this in Section 4.2 footnote 9.
>
> >  Throughout the paper, the authors appeal to intuition concerning putative human performance on the benchmarks they consider (e.g. in considering the potential for human reasoners to show a discrepancy between hypothesis generation and rule application), but intuition is not always a reliable guide regarding human performance. It would be good to qualify these statements a bit more (or, to the extent possible, to include a direct comparison with human performance).
>
> Thanks for the feedback! We conducted more human studies to provide a more comprehensive head-to-head comparison between the LM and humans. The additional human experiments include asking crowdworkers to (1) write rules for 50 randomly sampled tasks from List Functions and MiniARC (as a comparison to Table 1), (2) write rules for perturbed tasks with a hint indicating that examples may be incorrect (comparable to the dashed line in Figure 4a), and (3) write rules for perturbed tasks without hints (comparable to the bar Figure 4a). We included all details of these human studies in Appendix C.3. While these studies do not cover all the experiments conducted with LMs due to cost constraints, we hope they provide a better understanding of the behaviors of LMs and humans and motivate future research in this area.
>
> >  The 'task accuracy' measure seems designed to emphasize the consistency of the symbolic rule application. When looking at raw accuracy, the differences between symbolic and LLM rule application don't look nearly as large. I think this somewhat undermines the claim that their rule application abilities are so much worse than their hypothesis generation abilities.
>
> We consider “task accuracy” as an alternative metric as it captures a desirable property of an induction system: it should ideally consistently solve examples within a task. We use this metric to approximate the model’s true understanding of a task because a model is less likely to solve all examples without using the expected computation. Although the differences in raw accuracy are not as large as those in task accuracy, they are still significant for certain tasks (e.g. relative performance drop is around 27% for MiniSCAN and around 42% for MiniARC). In addition, we observe that more advanced prompting techniques, such as self-consistency prompting and zero-shot CoT prompting, also do not bridge the gap (see Appendix B.2 for new experiments). As we argue in Section 4.1, we do not claim that we should expect LMs to behave as effectively as symbolic interpreters, but often the gaps are still large enough to support the argument.

---

> > ### Author Response · Authors · 2023-11-19
> > **Response to Reviewer MmPX (2/n)**
> >
> > >  It should be emphasized more that miniARC, unlike the other tasks, is a distinctly visual task, in that it requires understanding visual concepts like 'objectness'. It is somewhat unsurprising that a text-only model would show special difficulty on such a task.
> >
> > Thanks for the suggestions! We added some discussions in Section 3 to better explain the results. Also added explanation in Appendix A dataset details to emphasize that we only use textual representations of the original visual grids.
> >
> > >  Do the authors have any explanation for why the hypothesis refinement approach achieves worse performance on miniARC relative to the baselines? Given the visual nature of the task, it's not surprising that it doesn't help much, but I was surprised that it actually seems to impair performance.
> >
> > Many tasks in MiniARC can be solved through pattern matching. In such cases, learning a simple dictionary that maps colors between seen examples and unseen examples may be sufficient (see the bottom example in Table 11 in Appendix B.2). LMs have been shown to be good at pattern matching (Mirchandani et al., 2023). However, generating a rule involves a higher level of abstraction, which could be more challenging for LMs. Importantly, in our setup, the interpreter does not have access to seen examples. Therefore, the rule must contain sufficient and accurate information for the interpreter to achieve strong performance when applying the rule. We explained this in the updated version in Section 3.
> >
> > Suvir Mirchandani, Fei Xia, Pete Florence, Brian Ichter, Danny Driess, Montserrat Gonzalez Arenas, Kanishka Rao, Dorsa Sadigh, Andy Zeng. Large Language Models as General Pattern Machines. 2023.
> >
> >
> > >  The familiarity analysis seems to confound two very different issues: 1) The presence of the same exact problems in the LLM pretraining data (a significant possibility, given the use of pre-existing datasets), and 2) the use of familiar vs. unfamiliar words (e.g. pseudowords). I think it's important to dissociate these concerns. This can be done by creating a new dataset with similar properties, e.g. by replacing the specific pseudowords and colors in miniSCAN (but maintaining the same general pseudoword -> color structure).
> >
> > We share similar concerns. For MiniSCAN, we did not use any existing examples, but instead generated our own examples to mitigate the potential data contamination and uncontrolled lexical exposure issues (see Appendix A for dataset details). The “pseudoword → color” follows the same setup in Lake et al. (2019), but uses different re-generated pseudowords for inputs. The “pseudoword → pseudoword” uses re-rengerated pseudowords for both input and output. We clarified this in the updated version.
> >
> > For the “pseudoword → color” setup, it is still possible that the output has been seen during pre-training. To eliminate this factor, we experiment a new setup where we replace the color names used in Lake et al. (2019) to other common color names. For instance, the “dax → RED” can become “dax → ORANGE”. We evaluate 100 tasks and show results below.
> >
> > |                       Setup | Raw Acc. | Task Acc. |
> > |-----------------------------|----------|-----------|
> > | pseudoword → original color |     93.3 |      85.0 |
> > |      pseudoword → new color |     97.1 |      95.0 |
> > |     pseudoword → pseudoword |     86.6 |      71.0 |
> >
> > The “pseudoword → original color” and “pseudoword → new color” setups both achieve higher accuracy compared to the “pseudoword → pseudoword” setup. This is consistent with our claim that LMs’ performance drops when the output representation deviates from their pre-training distribution.

---

> > > ### Author Response · Authors · 2023-11-19
> > > **Response to Reviewer MmPX (3/n)**
> > >
> > > >  Is the interaction between iid vs. ood and IO vs. hypothesis refinement (figure 2) statistically significant? Also, the text describes the hypothesis refinement results as demonstrating superior robustness to this ood setting, but for miniARC the ood accuracy is actually higher for the IO baseline (even though the difference between iid and ood performance is larger for IO). This should be clarified in the text.
> > >
> > > We clarified hypothesis refinement results (emphasizing the raw accuracy of IO prompting is still better than hypothesis refinement on MiniARC) in Section 3.
> > >
> > > We added two additional experiments. First, we run paired T-tests on IID vs OOD examples for IO and hypothesis refinement (250 tasks for List Functions and 130 tasks for MiniARC). We show the p-value below.
> > >
> > > |      | List Fns | MiniARC |
> > > |------|----------|---------|
> > > | IO   | 5.9e-19  | 5.4e-4  |
> > > | Rule | 4.2e-7   | 2.5e-2  |
> > >
> > > The difference between IID and OOD using IO prompting is much more statistically significant than rule prompting, suggesting that hypothesis refinement is more robust to OOD generalization.
> > >
> > > Additionally, we also run hypothesis refinement (T=3, N=5) for 3 runs and report the mean and standard deviation. We did not run multiple runs for IO prompting, since the results are deterministic when using greedy decoding. The results are shown below. The standard deviation is low across multiple runs.
> > >
> > > List Functions
> > > |  Run | IID Raw Acc. | OOD Raw Acc. | IID Task Acc. | OOD Task Acc. |
> > > |------|--------------|--------------|---------------|---------------|
> > > |    1 |         71.2 |         65.7 |          61.2 |          62.4 |
> > > |    2 |         67.5 |         63.9 |          58.4 |          59.2 |
> > > |    3 |         71.2 |         66.1 |          61.6 |          63.2 |
> > > | Mean |         70.0 |         65.2 |          60.4 |          61.6 |
> > > |  Std |          1.7 |          1.0 |           1.4 |           1.7 |
> > >
> > > MiniARC
> > > |  Run | IID Raw Acc. | OOD Raw Acc. | IID Task Acc. | OOD Task Acc. |
> > > |------|--------------|--------------|---------------|---------------|
> > > |    1 |         18.7 |         13.5 |          14.6 |          12.3 |
> > > |    2 |         17.2 |         12.3 |          12.3 |          10.8 |
> > > |    3 |         16.7 |         11.5 |          13.1 |          10.0 |
> > > | Mean |         17.2 |         12.3 |          13.1 |          11.0 |
> > > |  Std |          1.0 |          1.0 |           1.2 |           1.2 |
> > >
> > > >  Were the language model and human hypotheses systematically compared in any way, or only qualitatively inspected? Did the strength of the language model hypotheses correspond to performance on the task? Were there cases where the model performed well on the task despite providing unhuman-like hypotheses?
> > >
> > > In addition to qualitative evaluation, we ask human annotators to quantitatively rate between LM-induced rules and human-induced rules using two metrics: clarity and supportiveness. See Appendix C for details. For List Functions, where the LM achieves high accuracy, LM-induced rules and human-induced rules are comparably clear, but the former are sometimes less supportive. On MiniARC, where the LM performs poorly, we observe a significant performance gap between LM-induced rules and human-induced rules on both clarity and supportiveness. This seems to suggest that good performance is correlated with similarity with human-rules. We haven't found cases where unhuman-like hypotheses can still achieve very good performance in our experiments.

---

> > > > ### Author Response · Authors · 2023-11-19
> > > > **Response to Reviewer MmPX (4/n)**
> > > >
> > > > ### Response to minor comments
> > > >
> > > > >  What setting was used for the 'top p' parameter in GPT-4?
> > > >
> > > > We use default `top_p = 1`.
> > > >
> > > > >  It would be helpful to either include the variable names in Figure 1, or to have a separate figure illustrating the overall flow of the model with the corresponding variable names.
> > > >
> > > > Thanks for your suggestion! Due to the space limit, we add a sentence in the caption of Figure 1 to clarify the two main hyperparameters (T, N) used in iterative refinement.
> > > >
> > > > >  It would be good to say a bit more about how this work differs from Wang et al (2023). This is concurrent work, so there is no concern about novelty, but it would still be useful to have more discussion of the relationship.
> > > >
> > > > Our paper shares a similar spirit with Wang et al. (2023) in exploring the inductive reasoning capabilities of LMs. However, we focus on understanding both the potentials and limitations of LMs. For this reason, our work differs from Wang et al. (2023) from several perspectives.
> > > >
> > > > - *Methodology-wise*, we explore various domains and different types of interpreters to validate the effectiveness of iterative hypothesis refinement. We minimized task-specific heuristics (e.g. we did not explicitly explain useful visual concepts like “objectness” in MiniARC) and did not exhaustively tune hyper-parameters. We expect better performance by exploring alternative prompt templates and tuning hyper-parameters, as suggested by Wang et al. (2023).
> > > > - *Analysis-wise*, we offer more exploration of the limitations of LMs and identify several counterintuitive behaviors of LMs in Section 4. We also conduct human studies (Appendix C) to contrast the inductive reasoning capabilities of LMs with those of humans.
> > > >
> > > > We discuss the relationship between our work and Wang et al. (2023) at the end of Section 1 and in the related work section. We also point out in Section 5 that some techniques used in Wang et al. (2023) could potentially further improve performance.
> > > >
> > > > >  I found the description of the approach somewhat confusing. Based on the abstract and intro, I was expecting that the hypotheses would be articulated in natural language, and this would somehow be translated into code which is then symbolically executed. It is explained later on (in section 2.2) that the LM also carries out this translation step for list functions and miniARC, but it would be good to provide some hint that this is the case earlier in the section describing the approach. My understanding is that for the other tasks, the LM is prompted so as to ensure hypotheses in a particular format, which can then be automatically parsed, is that correct? It would be helpful to clarify this (in section 2 it says that the hypotheses are 'constrained', but it's not immediately clear what that means).
> > > >
> > > > Yes, for ACRE and MiniSCAN, the LM is prompted to generate hypotheses in a particular format. We clarified this in the introduction to emphasize that the hypothesis could be either free-form (in which case we use an LM to translate the hypothesis to a format interpretable by the symbolic interpreter) or constrained (in which case we can directly pass it to the interpreter).
> > > >
> > > > Additionally, for List Functions and MiniARC, we also evaluate directly generating Python hypotheses (see Appendix B.2. Representation of Hypothesis). As shown in Table 9, using programming language hypotheses achieve comparable performance to natural language hypothese, suggesting that this representation can be a promising alternative for these tasks.
> > > >
> > > > >  I found this sentence, 'However, they also behave as puzzling inductive reasoners, showing notable performance gaps in rule induction (i.e., identifying plausible rules)…' to be confusing, because it seems to say that LLMs are bad at proposing rules, even though it was just stated that they are good at this. This also seems misaligned with the results. It seems like this sentence should instead emphasize the rule application specifically (though, as mentioned above, it's not clear how significant this discrepancy really is).
> > > >
> > > > We intend to highlight that there are significant performance gaps between rule induction (good) and rule application (poor). In other words, LMs are much more effective at generating meaningful hypotheses than at applying the rules they propose. We hope this clarifies the confusion.

---

> > > > > ### Comment · Reviewer_MmPX · 2023-11-22
> > > > > **Response to rebuttal**
> > > > >
> > > > > Thank you to the authors for this extremely thorough rebuttal and revision. I am happy to raise my score to an 8 based on these updates.
> > > > >
> > > > > I do think that it would still be worth emphasizing that the intuition regarding the discrepancy between rule generation and rule application in human reasoners is, as far as I can tell, unsupported by human behavioral data (unless I missed it, the newly added behavioral results do not address rule application in human subjects). There is also no human behavioral data on the 'task accuracy' measure, which I believe may give a somewhat exaggerated view of performance differences between LM and symbolic rule application. So even though the difference between these approaches may seem quite large and at odds with human intuition, I don't think we can say for sure how human participants would perform on this metric. I don't think this in any way undermines the results of this study, but I think a more qualified statement about intuitions regarding human performance would be helpful.
> > > > >
> > > > > Regarding the sentence in the abstract that I found confusing, I think it would be clearer if it were rephrased to read: 'However, they also behave as puzzling inductive reasoners, showing notable performance gaps *between* rule induction (i.e., identifying plausible rules) and rule application (i.e., applying proposed rules to instances)...'

---

> > > > > > ### Author Response · Authors · 2023-11-22
> > > > > > **Thank you for your feedback**
> > > > > >
> > > > > > Thank you so much for your super useful feedback! We will emphasize your point in the revision and update the abstract based on your suggestion.

---

### Official Review · Reviewer_6Upx · 2023-11-01

**Soundness:** 4 excellent
**Presentation:** 4 excellent
**Contribution:** 3 good
**Rating:** 8
**Confidence:** 5

**Summary:**

The paper explores the inductive reasoning capabilities of large language models (LLMs) through iterative hypothesis refinement. The key ideas are:

- Inductive reasoning involves proposing hypotheses to explain observations, selecting the best hypothesis, and refining it based on new examples. This process mirrors
human inductive reasoning.
- The authors test LLMs on this through:
    1. Using the LLM to propose rule hypotheses based on examples
    2. Testing the rules using symbolic interpreters or LLMs as rule appliers on new examples
    3. Providing feedback to the LLM to further refine the rules
- Experiments on 4 datasets show LLMs are phenomenal at proposing plausible hypotheses when combined with symbolic interpreters. Iterative refinement significantly improves
performance.
- However, LLMs display counter-intuitive inductive behaviors compared to humans:
    - They struggle to apply their own proposed rules
    - They are brittle to minor perturbations in examples
    - Their induced rules differ in content and form from human-proposed rules

**Strengths:**

- Well motivated, clear and flows well. I really enjoyed reading the paper.
- The paper tackles an important problem in reasoning, reasoning inductively by proposing hypotheses.
- The domains are well defined and the content is diverse.
- The human experiments are insightful - comparing induced rules reveals qualitative gaps between LLMs and human reasoning.
- The paper makes an important contribution in carefully evaluating both strengths and weaknesses of LLMs for inductive reasoning.
- The analysis is thorough, spanning different models, datasets, and evaluations.
- The limitations, scope and results are clearly defined and discussed.

Overall, this is a clearly written, rigorous, and impactful study that advances our understanding of inductive reasoning in LLMs. The paradoxical findings are intriguing and point to promising future directions.

**Weaknesses:**

- An analysis of the complexity of the rules used to generate the data would be interesting. Comparing the complexity of the hypothesis across tasks and domains might give some insight into the model performance.
- Similarly, the complexity of the human induced and LLM induced rules might be interesting to analyze.
- How were the number of examples seen by the model chosen across domains? What is the minimum number of examples needed to learn a rule?
- An open source model would make the evaluations more comprehensive.
- A separate evaluation for LLMs as symbolic interpreters of rules would help tease apart the rule-proposing / application componenets more. More on complexity: LMs might be bad appliers of complex rules.
- Can LLMs apply rules induced by humans?
- Is there a change in the types of rules induced if the prompt is changed to encourage communication (since this was what humans seemed to do)? Change prompt to emphasize communication?
- MiniAC→MiniARC: 4.3 para1 line 3

**Questions:**

I have specified the questions/ suggestions in the weaknesses section.

---

> ### Author Response · Authors · 2023-11-19
> **Response to Reviewer 6Upx (1/2)**
>
> Thank you for your valuable feedback and insightful suggestions! We appreciate that you found our paper interesting and insightful.
>
> >  An analysis of the complexity of the rules used to generate the data would be interesting. Comparing the complexity of the hypothesis across tasks and domains might give some insight into the model performance.
>
> It’s definitely very interesting to analyze the relationship between the complexity of the rules and model performance. However, it is challenging to determine a good measure of complexity. As a preliminary experiment, we investigate how rule complexity affects model performance using List Functions, as its data was generated by programs. For each task, we collect its Python Lambda expression and analyze its complexity using abstract syntax tree (AST), similar to Bi et al. (2023). Note that the original List Function dataset uses a domain-specific language (DSL). This DSL defines symbols for basic list operations following a Lisp-like syntax. For instance, a list `[1, 2, 3]` is represented as `(cons 1 (cons 2 (singleton 3)))`. We did not consider this DSL for our analysis, as we hypothesize that Python is better represented in LMs' pre-training data than this DSL. Analyzing Python expressions, therefore, could provide a better estimate of rule complexity.
>
> We consider both primitive and structural complexities and investigate their correlations with model performance. The former indicates the learnability of individual primitive operations, while the latter reflects the difficulty of the compositional structures of the rule, which we approximate using the number of nodes and tree depth of the AST.
>
> We first examine how different primitive operations contribute to rule induction. We train a logistic regression model to predict LM's task accuracy on a new instance, using the counts of different types of nodes as features. The logistic regression models achieve an averaged test accuracy of 68.4% with a small number of features, suggesting that the model performance is somewhat predictable based on the primitives of the rule. We use the coefficient as an approximation of the learnability of the primitive operations.  These coefficients vary across different primitives, with some operations such as `In` being easier to learn than others like `Sub`.
>
> For structure complexity, we measure the Spearman correlations between test accuracy and the structure complexity measurements. The number of nodes has a correlation of -0.31, and tree depth has a correlation of -0.37. Tasks with high accuracy generally have a relatively small number of nodes and tree depth. However, some programs with low structural complexity also have low accuracy, indicating that other factors, such as the aforementioned primitive complexity, should also be considered.
>
> Zhen Bi, Ningyu Zhang, Yinuo Jiang, Shumin Deng, Guozhou Zheng, Huajun Chen. When Do Program-of-Thoughts Work for Reasoning?
>
> >  Similarly, the complexity of the human induced and LLM induced rules might be interesting to analyze.
>
> Since the human-induced and LM-induced rules are often in free-form natural language, it is even more challenging to measure their complexity. Measures such as minimum description length and the number of unique n-grams could be potential solutions. However, it is unclear whether these measures correlate with the complexity of rules in our setup. We agree that this is a very interesting direction, but it is beyond the scope of this paper. Therefore, we leave this as a direction for future work.

---

> > ### Author Response · Authors · 2023-11-19
> > **Response to Reviewer 6Upx (2/2)**
> >
> > >  How were the number of examples seen by the model chosen across domains? What is the minimum number of examples needed to learn a rule?
> >
> > We leverage the existing datasets for our experiments, therefore the number of seen examples for each dataset is given, except for List Functions where we only use 8-shot examples. Our preliminary experiments on List Functions suggest that increasing the number of seen examples does not improve performance. Since our tasks are relatively self-defined, a small number of seen examples is generally sufficient. Finding the minimum number of examples necessary for induction is definitely an interesting and ongoing research direction! However, as we noted in footnote 11, this is out of the scope of this paper.
> >
> > >  An open source model would make the evaluations more comprehensive.
> >
> > We included LLaMA2-70B results in Appendix B.1.
> >
> > >  A separate evaluation for LLMs as symbolic interpreters of rules would help tease apart the rule-proposing / application componenets more. More on complexity: LMs might be bad appliers of complex rules.
> >
> > We include this analysis in Section 4.1. In Figure 3, we compare the accuracy when applying the rules using symbolic interpreters or the LM itself as the interpreter. We observe a consistent performance drop when using the LM interpreter instead of the symbolic interpreter. This suggests that while LMs are effective at proposing rules, they may not be able to apply their own proposed rules.
> >
> > >  Can LLMs apply rules induced by humans?
> >
> > Thanks for the suggestion! Our focus is to evaluate inductive reasoning capabilities of LMs, therefore we did not evaluate whether LMs can apply rules induced by humans. Wang et al. (2023) show that human-written hypotheses achieve the strongest performance on ARC, suggesting that human-induced rules might be helpful. Note that their setup still only uses the LM to translate hypotheses and relies on an external interpreter to apply the rules. It’s definitely super interesting to explore this direction, but since it’s beyond the scope of this paper, we leave it as a potential direction for future work.
> >
> > Ruocheng Wang, Eric Zelikman, Gabriel Poesia, Yewen Pu, Nick Haber, Noah D. Goodman. Hypothesis Search: Inductive Reasoning with Language Models.
> >
> > >  Is there a change in the types of rules induced if the prompt is changed to encourage communication (since this was what humans seemed to do)? Change prompt to emphasize communication?
> >
> > First of all, to investigate whether other prompts would lead to different behaviors, we evaluate an alternative hypothesis generation prompt that includes task-specific heuristics (see Appendix B.2, Task-specific Heuristics). Specifically, we use MiniARC and add a detailed task description in the prompt (Table 10). This does help LMs to generate more human-readable rules (Table 11) for a fraction of examples, but these rules are still generally distinguishable from those induced by humans. Additionally, we did not observe performance improvement by imposing these constraints. While it is possible that other prompts could encourage LMs to communicate more, this would require additional prompt engineering and is beyond the scope of this paper. We mention this as a potential direction for future work.
> >
> > Secondly, for a fair comparison, we also did not instruct annotators to use any communication strategies nor did we provide any constraints or hints on the types of rules (see annotation interface in Figure 8). Humans appear to naturally use pragmatic communication strategies. Our point is that, without any specific instruction, LMs demonstrate inductive reasoning behaviors that are different from those of humans. Emphasizing communication could certainly be useful from a practical standpoint, but it is not our main focus in understanding the limitations of LMs in inductive reasoning.

---

### Official Review · Reviewer_bJW9 · 2023-11-01

**Soundness:** 3 good
**Presentation:** 4 excellent
**Contribution:** 3 good
**Rating:** 8
**Confidence:** 4

**Summary:**

This paper investigates the inductive reasoning capacities of language models on a set of tasks, in terms of hypothesis proposal, selection, and refinement and then analyzes how the hypotheses differ from human ones.

**Strengths:**

Although the tasks are somewhat toy, the paper demonstrates its claims, is well-written, and is relatively comprehensive. They perform a novel analysis of the kinds of hypotheses and the model's ability to apply them. This is (in my view) a clear contribution, and I have no substantial criticisms.

**Weaknesses:**

A few of the experiment setups feel a bit contrived - for example, randomly perturbing a set of items in a small set of experiments, of course, makes the task harder for a language model since it also requires it to infer that the noise is noise and not itself a deterministic part of the rule. The section on familiarity of exemplars should also likely mention Dasgupta et al.'s "Language models show human-like content effects" there.

**Questions:**

I'm curious about how this paper squares with some results like that in the after-submission-deadline "Large Language Models Cannot Self-Correct Reasoning Yet" from Huang et al. (2023). The point there was that language models, given the opportunity to revise their reasoning, will often make it worse. In practice, did you see this when refining hypotheses? It would be interesting to see how the number of revisions affects performance, similar to Table 2 in the concurrent "Hypothesis Search" paper from Wang et al. I see there is some version of this in this paper's Table 2, but given the emphasis that this paper places of hypothesis refinement, I'd expect a bit more detail. Especially given the self-consistency, it would be valuable to understand the tradeoff between more attempts and more revisions.

---

> ### Author Response · Authors · 2023-11-19
> **Response to Reviewer bJW9**
>
> Thank you for your valuable feedback! We appreciate your support for our paper.
>
>
> ### Response to weakness
>
> >  A few of the experiment setups feel a bit contrived - for example, randomly perturbing a set of items in a small set of experiments, of course, makes the task harder for a language model since it also requires it to infer that the noise is noise and not itself a deterministic part of the rule.
>
> To better calibrate model performance, we conducted human study using the same setup and added results in Appendix C.3. We observe that both the LM and humans perform worse on tasks with noisy examples. However, the relative performance drop of the LM is more significant. We clarified this in Section 4.2 footnote 9.
>
> >  The section on familiarity of exemplars should also likely mention Dasgupta et al.'s "Language models show human-like content effects" there.
>
> Thanks for the pointer! We discussed this paper in Section 4.2 footnote 10.
>
> ### Response to questions
>
> >   In practice, did you see this when refining hypotheses? It would be interesting to see how the number of revisions affects performance, similar to Table 2 in the concurrent "Hypothesis Search" paper from Wang et al. I see there is some version of this in this paper's Table 2, but given the emphasis that this paper places of hypothesis refinement, I'd expect a bit more detail. Especially given the self-consistency, it would be valuable to understand the tradeoff between more attempts and more revisions.
>
> Huang et al. (2023) evaluates *intrinsic* self-correction without external feedback, which is similar to Self-Refine. As shown in Table 1, we similarly found an iterative approach without external feedback is insufficient. However, we did observe performance improvement over iterations for all datasets when coupled with external interpreters. The results are shown below. See Figure 5 in Appendix B.3 for a better visualization. This is consistent with the findings in Huang et al. (2023), as they discussed in Section 5 “Leveraging external feedback for correction”.
>
> Raw Accuracy
> | Iteration |    1 |    2 |    3 |
> |-----------|-----|-----|-----|
> | ACRE      | 79.8 | 82.3 | 82.5 |
> | MiniSCAN  | 86.6 | 92.9 | 93.3 |
> | List Fns  | 62.4 | 68.3 | 71.2 |
> | MiniARC   | 12.8 | 15.4 | 18.7 |
>
> Task Accuracy
> | Iteration |    1 |    2 |    3 |
> |-----------|-----|-----|-----|
> | ACRE      | 48.0 | 56.0 | 59.0 |
> | MiniSCAN  | 70.0 | 84.0 | 85.0 |
> | List Fns  | 52.4 | 58.4 | 61.2 |
> | MiniARC   |  9.2 | 12.3 | 14.6 |
>
> However, as we mentioned at the end of Section 4.3, qualitatively, in some cases, LMs tend to make minor modifications rather than starting from entirely new hypotheses. Therefore, in cases where the initial hypothesis is completely off, iterative refinement tends to be less effective.
>
> We agree that it is certainly interesting to investigate the tradeoff between the number of hypotheses (N) and the maximum number of iterations (T). We did preliminary exploration (in Table 1) to demonstrate the correlations between model performance and these two hyperparameters. For tasks where LMs achieve strong performance, such as ACRE and MiniSCAN, a limited number of iterations is already effective. For tasks like MiniARC, where LMs perform poorly, the trends remain positive after the maximum number of iterations.
>
> Note that the maximum number of iterations is bounded by the context lengths of LMs. Empirically, we find it is infeasible to further increase T. An alternative could be tracking only the latest K iterations. Additionally, increasing N leads to more API calls, which can be computationally expensive. A summarization strategy, as suggested in Wang et al. (2023) might offer a solution. Therefore, identifying the best tradeoff between more attempts and more revisions could require more method-wise exploration, which is beyond the scope of this paper. We hope our experiments motivate future work along this direction.

---

> > ### Comment · Reviewer_bJW9 · 2023-11-22
> >
> > Thank you for your response! I will keep my score as is.

---

### Official Review · Reviewer_EAiR · 2023-11-01

**Soundness:** 3 good
**Presentation:** 3 good
**Contribution:** 3 good
**Rating:** 8
**Confidence:** 3

**Summary:**

The paper focuses on evaluating the inductive reasoning capabilities of large language models. Given input examples, the authors propose a three-stage process that first asks LLMs to propose hypotheses about the task, and then use a domain-specific interpreter to evaluate hypotheses on input examples, finally, hypotheses that pass most input examples are used to apply to unseen examples for testing.  The authors also propose to leverage interpreter results of hypotheses as feedback to refine the hypotheses. Experiments show that this approach significantly boosts the performance of LLMs on 4 inductive reasoning datasets. The authors then show various differences between humans and LLMs through additional experiments such as asking LLMs to apply rules without interpreters and perturbing part of input examples. These experiments demonstrate the behavior difference between humans and LLMs on inductive reasoning tasks.

**Strengths:**

- The author proposes an effective approach that disentangles the inductive reasoning task into the process of proposing a hypothesis and interpreting the hypothesis that shows strong performance compared with recent approaches that use various types of prompting without external interpreters.
- The proposed method is validated on multiple large language models by comprehensive experiments on 4 datasets of different domains, showing the generalizability of the method.

**Weaknesses:**

My concern mainly lies in Section 4:

- For the example perturbation experiment in section 4.2, there are no studies on how well humans can actually perform on perturbed tasks. It is hard to judge how big the performance drop of LMs is compared with humans.
- Experiments in 4.1 and 4.2 are conducted with simple prompting which may not be the most effective method to elicit this type of reasoning from the model.
- The generalizability of the findings in 4.3 is doubtful because only one type of prompt is used to generate rules from LM. According to the appendix, example rules on List Fn and MiniARC are generated from LM with a prompt that contains no format instruction. It is unknown whether LMs can generate human-like inductions with more guidance or provided with human-induced rules as few shot examples.

Overall, I believe the main contribution of the paper is an effective inductive reasoning pipeline using LMs. So these are not significant flaws of the paper. So I still recommend acceptance. However, I strongly encourage authors to provide more rigorous evidence when making claims in Section 4.

**Questions:**

- What are the prompts used to generate the Python program from rules for List Functions and MiniARC?
- In Table 2, on MiniScan, T=3, N=1 yields higher raw accuracy and task accuracy than T=3, N=5, which means that fewer hypotheses considered lead to worse performance. Is there an explanation for this?
- In Section 2, the authors claim to evaluate models on “OOD” examples by generating longer or larger examples than those in the original datasets. This is a bit confusing. Are authors actually fixing the seen examples and only changing the unseen examples for testing?
- When asking LMs to write Python programs given the hypothesis, are the seen examples also provided in the prompt?

---

> ### Author Response · Authors · 2023-11-19
> **Response to Reviewer EAiR (1/2)**
>
> Thank you for your valuable feedback! We appreciate that you found our approach effective and our experiments comprehensive.
>
>
> ### Response to weakness
>
> >  For the example perturbation experiment in section 4.2, there are no studies on how well humans can actually perform on perturbed tasks. It is hard to judge how big the performance drop of LMs is compared with humans.
>
> Thanks for the feedback! We conducted human study using the same setup and added results in Appendix C.3. We observe that both the LM and humans perform worse on tasks with noisy examples. However, the relative performance drop of the LM is more significant. We have clarified this in Section 4.2 footnote 9.
>
> >  Experiments in 4.1 and 4.2 are conducted with simple prompting which may not be the most effective method to elicit this type of reasoning from the model.
>
> For section 4.1, we consider two alternative prompting techniques for rule application: self-consistency prompting (SC) and zero-shot chain-of-thought (0-CoT). The results are shown below. We do not observe significant performance differences across these methods, except on ACRE, where 0-CoT underperforms other methods in task accuracy. We included detailed results and discussions in Appendix B.2 and clarified this in Section 4.1 Our results show the rule application performance is consistent across various prompting methods. It would be interesting to explore more advanced prompting techniques for our setting, but that is out of scope of this paper.
>
> Raw Accuracy
> | Method   | ACRE | MiniSCAN | List Fns | MiniARC |
> |----------|------|----------|----------|---------|
> | Standard | 77.8 | 67.6     | 65.8     | 10.8    |
> | 0-CoT    | 73.2 | 65.5     | 61.2     | 12.1    |
> | SC (N=5) | 77.0 | 67.5     | 66.3     | 9.7     |
>
> Task Accuracy
> | Method   | ACRE | MiniSCAN | List Fns | MiniARC |
> |----------|------|----------|----------|---------|
> | Standard | 47.0 | 0.0      | 50.0     | 5.4     |
> | 0-CoT    | 25.0 | 0.0      | 48.4     | 6.9     |
> | SC (N=5) | 46.0 | 0.0      | 50.8     | 4.6     |
>
> For section 4.2, all perturbation experiments use iterative hypothesis refinement (T=3, N=5), which has the strongest performance in our main experiments. We clarified this in the main text and included results using other models and configurations in Appendix B.2.
>
> >  The generalizability of the findings in 4.3 is doubtful because only one type of prompt is used to generate rules from LM. According to the appendix, example rules on List Fn and MiniARC are generated from LM with a prompt that contains no format instruction. It is unknown whether LMs can generate human-like inductions with more guidance or provided with human-induced rules as few-shot examples.
>
> We did not introduce task-specific format instructions for List Functions and MiniARC, as they cover a wide range of concepts, making it challenging to determine the most optimal constraints. To investigate whether other prompts would lead to different behaviors, we evaluate an alternative hypothesis generation prompt that includes task-specific heuristics (see Appendix B.2, Task-specific Heuristics). Specifically, we use MiniARC and add a detailed task description in the prompt (Table 10). This does help LMs to generate more human-readable rules (Table 11) for a fraction of examples, but these rules are still generally distinguishable from those induced by humans. Additionally, we did not observe performance improvement by imposing these constraints. It is certainly possible that other guidance or few-shot examples could encourage LMs to generate human-like rules. However, the former requires more prompt engineering, and the latter requires additional human annotations, which are beyond the scope of this paper. We mention these alternatives as potential directions for future work.

---

> > ### Author Response · Authors · 2023-11-19
> > **Response to Reviewer EAiR (2/2)**
> >
> > ### Response to questions
> >
> > >  What are the prompts used to generate the Python program from rules for List Functions and MiniARC?
> >
> > We use the following prompt template:
> >
> > “You are an expert Python programmer. Write a Python function `fn` for the following rule. {Example description}
> >
> > Rule: {Rule}”
> >
> > The {Example description} describes the input and output types, e.g. a list of integers (List Functions) or a 2D grid of integers (for MiniARC). We show the hypothesis translation prompt in Table 14 in Appendix D.
> >
> > >  In Table 2, on MiniScan, T=3, N=1 yields higher raw accuracy and task accuracy than T=3, N=5, which means that fewer hypotheses considered lead to worse performance. Is there an explanation for this?
> >
> > One possible explanation for these results could be the differences in sampling temperature. For N=1, we use greedy decoding (temperature = 0), and for N=5, the temperature is set to 0.7. An increased temperature might lead to more inaccurate hypotheses, similar to what is observed with SC prompting compared to IO prompting.
> >
> > More importantly, the MiniSCAN dataset is designed to evaluate compositional generalization: certain unseen examples require the novel composition of known concepts. The distribution shift between seen examples and unseen examples means that high accuracy on seen examples does not necessarily translate to high accuracy on unseen examples. For instance, the rule “dax fep → RED RED RED” is less preferable, even if it perfectly explains the given example. Since iterative refinement only uses accuracy over seen examples as the scoring function, it might overfit to seen examples and select hypotheses that are less generalizable. We clarified this in Section 3.
> >
> > >  In Section 2, the authors claim to evaluate models on “OOD” examples by generating longer or larger examples than those in the original datasets. This is a bit confusing. Are authors actually fixing the seen examples and only changing the unseen examples for testing?
> >
> > Yes. We clarified this in the updated version.
> >
> > >  When asking LMs to write Python programs given the hypothesis, are the seen examples also provided in the prompt?
> >
> > No, we do not provide examples. See Table 14 in Appendix D for hypothesis translation prompt.

---

> > > ### Comment · Reviewer_EAiR · 2023-11-23
> > >
> > > Thanks authors for the detailed response. I raise my rating to 8.

---

### Author Response · Authors · 2023-11-19
**General Response**

We thank all the reviewers for their thoughtful and valuable feedback! We have conducted additional experiments and revised the paper based on the reviews. We highlighted all changes in blue. We also included more results and analyses to make the paper more comprehensive. Additionally, we fixed a small bug during the revision, which leads to small changes in main results. However, the general results remain consistent. Below is a summary of main changes. Please let us know if you have further questions.

| Change                                                                     | Section                  | Related Reviewers                                 |
|----------------------------------------------------------------------------|--------------------------|---------------------------------------------------|
| Human performance on perturbed tasks in Section 4.2                        | Appendix C.3             | Reviewer EAiR, Reviewer bJW9, Reviewer MmPX  |
| Human evaluation on LM-induced rules and human-induced rules               | Appendix C.3             | Reviewer MmPX                                     |
| Human performance from existing studies                                    | Appendix C.1             | Reviewer MmPX                                     |
| Alternative prompts with task-specific heuristic                           | Appendix B.2             | Reviewer EAiR, Reviewer 6Upx                      |
| Other prompting techniques for rule application                            | Appendix B.2             | Reviewer EAiR                                     |
| Results using open source model LLaMA2-70B                                 | Appendix B.1             | Reviewer 6Upx                                     |
| Additional analysis of accuracy over iterations                            | Appendix B.3             | Reviewer bJW9                                     |
| Ablation on explicitly instructing LMs to take into account noisy examples | Section 4.2              | Reviewer EAiR, Reviewer bJW9, Reviewer MmPX |
| Alternative representation for hypothesis: natural language vs. Python     | Appendix B.2             | Reviewer MmPX                                     |
| Average cost for each task                                                 | Appendix B.4             |                                                   |
| Results of other models                                                    | Section 3 → Appendix B.1 |                                                   |

All additional experiments are included in the Appendix. The Appendix is organized as follows.

- Appendix A: dataset details.
- Appendix B: additional results and ablations.
- Appendix C: human studies.
- Appendix D: prompts and examples.

---

### Meta-Review · Area_Chair_BVTR · 2023-12-03

**Metareview:**

The paper This investigates the inductive reasoning capabilities of large language models. The results suggest that they are often good at proposing hypotheses, but not so good at applying the proposed rules. Then a  neurosymbolic/toolformer approach is proposed, improving performance on multiple inductive reasoning tasks. All reviewers agree that this is solid work.

**Justification For Why Not Higher Score:**

N/A

**Justification For Why Not Lower Score:**

The topic is very important.

---

### Decision · Program_Chairs · 2024-01-16

Accept (oral)